# Scissorhands: Exploiting the Persistence of Importance Hypothesis for LLM KV Cache Compression at Test Time

**Zichang Liu**
Department of Computer Science
Rice University
zichangliu@rice.edu

**Aditya Desai**
Department of Computer Science
Rice University
Aditya.P.Desai@rice.edu

**Fangshuo Liao**
Department of Computer Science
Rice University
Fangshuo.Liao@rice.edu

**Weitao Wang**
Department of Computer Science
Rice University
wtwang@rice.edu

**Victor Xie**
Department of Computer Science
Rice University
vyx2@rice.edu

**Zhaozhuo Xu**
Department of Computer Science
Stevens Institute of Technology
zxu79@stevens.edu

**Anastasios Kyrillidis**
Department of Computer Science
Rice University
anastasios@rice.edu

**Anshumali Shrivastava**
Department of Computer Science
Rice University & ThirdAI Corp.
anshumali@rice.edu

## Abstract

Large language models(LLMs) have sparked a new wave of exciting AI applications. Hosting these models at scale requires significant memory resources. One crucial memory bottleneck for the deployment stems from the context window. It is commonly recognized that model weights are memory hungry; however, the size of key-value embedding stored during the generation process (KV cache) can easily surpass the model size. The enormous size of the KV cache puts constraints on the inference batch size, which is crucial for high throughput inference workload. Inspired by an interesting observation of the attention scores, we hypothesize *the persistence of importance*: only pivotal tokens, which had a substantial influence at one step, will significantly influence future generations. Based on our empirical verification and theoretical analysis around this hypothesis, we propose SCISSORHANDS, a system that maintains the memory usage of KV cache under a fixed budget without finetuning the model. We validate that SCISSORHANDS reduces the inference memory usage of the KV cache by up to $5\times$ without compromising model quality. We further demonstrate that SCISSORHANDS can be combined with 4-bit quantization for further compression

37th Conference on Neural Information Processing Systems (NeurIPS 2023).

# 1 Introduction

Large language models(LLMs), trained on immense amounts of text data, have demonstrated an incredible ability to generate text that is both logically connected and contextually relevant [1–5]. LLM inference follows an autoregressive fashion, generating one token at each step conditioned on the previous steps. At each step, the key-value embedding in attention is stored in memory to avoid repetitive key-value projection computation at future steps. Unfortunately, the memory of the key-value cache( KV cache), including prompts and previously generated tokens, can be surprisingly large. Using OPT-175B as an example, the impressive 175 billion parameters consume around 325 GB of memory. At the same time, at batch size 128 and sequence length 2048, the KV cache requires around 950 GB of memory, three times larger than the model weights. Considering that 8 Nvidia A100-80GB offers 640GB GPU memory, the memory usage of the KV cache is truly concerning.

LLMs are typically deployed on fixed memory hardware, and the size of model weights is also fixed once deployed. Apart from a small memory buffer typically reserved for communication and computation, the rest of the available memory is for the KV cache. The size of the KV cache depends on batch size, sequence length, and model dimension. Thus, at a given inference sequence length, compression in the KV cache memory translates almost linearly into an increase in the batch size. And any increase in batch size is significant for high-throughput inference systems [6, 7].

Quantization and sparsity approaches [8–14] have been studied in LLMs to reduce the model sizes. However, compressing the KV cache remains an open but challenging problem. First, training models at the scale of hundreds of billions of parameters on a large amount of data is prohibitively expensive. Thus, an ideal compression algorithm should be applicable without training. Second, emerging applications such as dialogue systems require an extremely long context window. The maximum sequence length of LLMs is growing to over 32K [15]. The size of the KV cache also grows linearly with sequence length. For scalability, an ideal compression algorithm should reduce the memory from the sequence length dimension. At last, compression should preserve LLMs' quality and in-context learning ability.

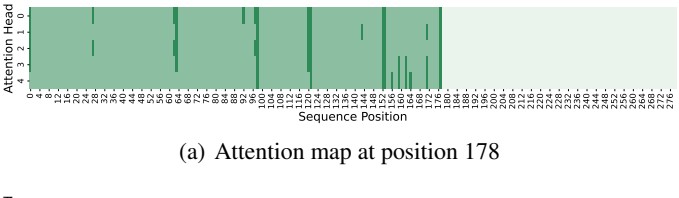

(a) Attention map at position 178

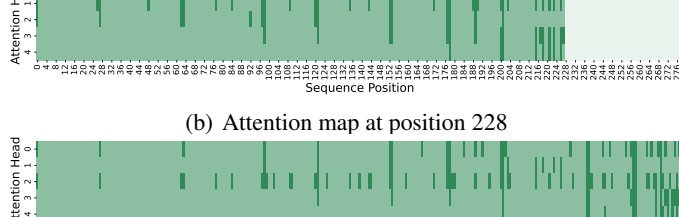

(b) Attention map at position 228

(c) Attention map at position 278

Figure 1: **Repetitive Attention Pattern**. We plot the attention map at three token positions in a sentence. Only five attention heads are plotted for a clearer presentation. We discretize the attention score such that the high score is dark green, and the low score is light green. In Figure 1(a), the token at position 178 pays heavy attention to positions 27, 63, 98, etc. This pattern is also present in the attention maps of position 228 and position 278.

We go beyond the traditional model compression techniques to achieve such demanding requirements. We envision that not all tokens must be stored in memory for LLM to understand the context. Just like humans can skim through an article and grasp the main idea, LLMs may also be able to skim and comprehend. It is commonly observed that the attention score from one token follows a strong power law distribution [16–20], meaning that one token will only heavily attend to a small number of tokens. More importantly, we observe **Repetitive Attention Pattern** from different tokens in the sequence in a trained LLM( Figure 1). Certain tokens are more important throughout the paragraph. Specifically, for two different tokens, there are similarities between which tokens they are heavily attending to and similarities between which tokens they are ignoring.

Inspired by the above observation, we articulate the **Persistence of Importance Hypothesis:** *Only pivotal tokens, which had a substantial influence at one previous step, will have a significant influence at a future step.* This hypothesis, if true, suggests that it is possible to foresee which token is likely to be important for future generations. Fortunately, we empirically verify that later tokens in the

sentence mostly only attend to tokens that were heavily attended from the early tokens in a sentence. And the overlapping ratio is surprisingly high, over 90% in most of the transformer layers (Figure 2).

Based on the above two findings, we present SCISSORHANDS that exploits the *persistence of importance hypothesis* to realize LLM inference with a compressed KV cache. In Section 4, we present an efficient algorithm such that the size of KV cache is always less than a predetermined budget. A theoretical guarantee justifies that such a compressed KV cache can approximate the attention output. In Section 5, we empirically evaluate SCISSORHANDS and show that SCISSORHANDS reduces the memory usage of KV cache $2-5\times$ without compromising model quality. Reduction in the KV cache can directly result in a larger batch size. Further, we adopt quantization and show its compatibility with SCISSORHANDS.

## 2 Problem Description and Related Work

This paper considers the LLM inference workflow, specifically focusing on the memory usage for storing the keys and values in attention. Let $d$ be the hidden dimension of the model, $b$ be the batch size, and $p$ be the length of prompt sentences. We are given the trained model weights, $W_K^i \in \mathbb{R}^{d \times d}$, $W_V^i \in \mathbb{R}^{d \times d}$ for the key and value projection matrix at the $i^{th}$ transformer layer.

The standard LLM inference consists of two stages: prompting and token generation. In the prompting stage, the model takes the prompt sentences as the input, and the key/value embedding in attention is stored as a cache to reduce repetitive computation. Denote $x_{\text{prompt}}^i = [x_1^i, ..., x_p^i], x_{\text{prompt}}^i \in \mathbb{R}^{b \times p \times d}$ as the input to attention at the $i^{th}$ transformer layer. Denote the key cache and value cache at layer $i$ as $\mathcal{K}^i, \mathcal{V}^i \in \mathbb{R}^{b \times p \times d}, \mathcal{K}_0^i = x_{\text{prompt}}^i W_K^i, \mathcal{V}_0^i = x_{\text{prompt}}^i W_V^i$.

In the generation stage, the model starts with the stored KV cache in the prompting stage and generates one token at each step. At each step, the KV cache gets updated. Given the input to attention at step $t$ in the $i^{th}$ transformer layer $x_t^i \in \mathbb{R}^{b \times 1 \times d}$. $\mathcal{K}_{t+1}^i = [\mathcal{K}_t^i, x_t^i W_K^i], \mathcal{V}_{t+1}^i = [\mathcal{V}_t^i, x_t^i W_V^i]$.

### 2.1 LLM Inference Memory Breakdown

In this section, we provide the memory consumption breakdown of LLMs. The memory footprint consists of three parts: model weights, KV cache, and activation buffer. The size of model weights depends on model configuration, such as the number of transformer layers and hidden size. The size of the KV cache depends on model configurations, sequence length, and batch size. The size of the activation buffer depends on parallelism strategy, model configurations, and implementation. The size of the activation buffer is considerably smaller than the previous two. As shown in Table 1, the size of the KV cache, $2.5\times$-$5\times$ larger than model weights, can quickly become the bottleneck in memory consumption. At the same time, much research has been spent on extending the length of the context window. GPT-4-32K can process up to 32,768 tokens [15]. Longer sequence length would make the KV cache memory problem even more severe.

Assuming LLM generates until its maximum sequence length, we summarize the maximum batch size before going out of GPU memory on a box of 8 A100 80GB GPU in Table 2. At the GPT-3 scale with a maximum sequence length of 2048, batch size cannot exceed 35 without offloading. Small batch size limits the model inference throughput.

### 2.2 Efficient Attention

Computing the attention matrix necessitates a time complexity of $O(n^2)$, where $n$ is the sequence length. As a result, a line of work has been proposed to mitigate the computation burden of the attention mechanism [16–20]. These approaches exploit low-rank or sparsification to approximate the attention output. Besides, [21] realized exact efficient attention with wall-clock speed by optimizing

Table 1: The memory consumption of model weights and KV cache for three different LLMs at batch size 128 and sequence length 2048 shows that the KV cache dominates the memory consumption.

| Model | # of Layer | Hidden Size | Weights (GB) | KV cache (GB) |
|-------|-----------|-------------|--------------|---------------|
| OPT-175B | 96 | 12288 | 325 | 1152 |
| LLaMA-65B | 80 | 8192 | 130 | 640 |
| BLOOM | 70 | 14336 | 352 | 950 |

Table 2: Maximum batch size before hitting out of memory on a box of 8 A100 80GB GPU when models are deployed with its maximum sequence length.

| Model | OPT-175B | LLaMA-65B | BLOOM |
|---|---|---|---|
| Maximum Batch Size | 34 | 102 | 36 |

the number of memory reads and writes. However, these approaches were evaluated mostly for training, focused on computation complexity, and did not address the KV-Cache memory usage introduced by auto-regressive language models.

Recently, there is active research attempting to apply quantization or pruning in LLM [8–14]. However, they mostly focus on reducing the size of model weights. Flexgen [7] applies quantization and sparsification to the KV cache; however, the memory of the KV cache is not reduced regarding sequence lengths. It stores the quantized KV cache for all tokens in CPU memory and loads all attention keys from CPU memory to compute attention scores. At the same time, methods such as Multi-Query-Attention(MQA) [22] change the attention design such that keys and values are shared across all attention heads. MQA requires training the model from scratch, while our works focus entirely on the inference stage.

## 3 The Persistence of Importance Hypothesis

We first present one interesting observation upon which the *persistence of importance hypothesis* is derived in Section 3.1. In Section 3.2, we discuss the hypothesis in detail with empirical verification. Then, in Section 3.3, we provide theoretical intuition on the reason behind such model behaviors.

### 3.1 Repetitive Attention Pattern.

**Observation.** We are interested in the attention score from the position $t$ over all the words that come before it in the sentence. In Figure 1, we provide three attention maps of a sentence randomly drawn from the Colossal Clean Crawled Corpus (C4) [23] using OPT-6B. Each attention map is a discretized attention score calculated at a random position. We consider a score larger than $\frac{1}{t}$ as significant as $\frac{1}{t}$ indicates an averaging mixing score. High attention scores are marked with dark green.

**Result.** High attention scores are observed at the same set of tokens from various positions in the sentence. In all three plots, we see dark green at sequence positions 27, 63, 98, 121, 152, and 177, suggesting that these tokens received high attention at all three positions. We observe similar model behavior at different transformer layers with different text inputs. More plots are in Appendix A.

**Implication.** Even though small differences exist, repetitive attention patterns are evident in the attention maps. There exist specific tokens that keep receiving high attention. Meanwhile, these attention maps show sparsity: only a few tokens have high attention scores.

### 3.2 The Persistence of Importance Hypothesis

The repetitive attention pattern suggests that specific tokens are influential throughout the sequence. A stricter claim is that these tokens are the only ones that could be significant for a future step. Thus, we articulate the *persistence of importance hypothesis*.

**The Persistence of Importance Hypothesis.** *With a trained autoregressive language model, only pivotal tokens, which had a substantial influence at one previous step, will have a significant influence at a future step.*

If true, this hypothesis indicates the possibility of foreseeing what information in the previous sequences could be vital for future steps. This hypothesis is trivial when pivotal tokens include all tokens in the entire sentences. However, a much more interesting case is when pivotal tokens are a subset of previous words. This would enable us to reduce the size of the KV cache by throwing away the embedding of non-important tokens.

**Pivotal Token.** One natural indication of a token's influence is the attention score. We consider a token pivotal for position $t$ if this token receives an attention score larger than threshold $\alpha$ from the token at position $t$. Let $S_t$ denote the set of pivotal tokens for position $t$. $S_{a \to b}$ denote the set of

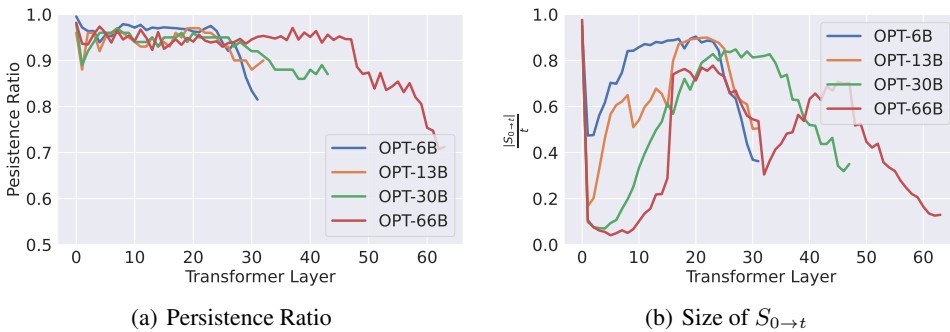

(a) Persistence Ratio     (b) Size of $S_{0 \to t}$

Figure 2: Persistence ratio and the corresponding size of the pivotal token set. The persistence ratio is over 95% in most layers, with decreases at the later layers. Meanwhile, the number of pivotal tokens is considerably smaller than the sequence length. This suggests that the pivotal tokens of later half sentences are almost all included in the set of first halves.

pivotal tokens for every position from $a$ to $b$.

$$S_{a \to b} = \cup_{t=a}^{t=b} S_t$$

**Verification.** We measure *persistence ratio* as an empirical test the hypothesis. *Persistence ratio* measures how many tokens in the pivotal token sets of the later part of the sentence are also in the pivotal token sets of the initial part of the sentence. Let $l$ denote the length of the sentence. We record $S_{1 \to t} \in \{x_1, ...x_t\}$, tokens in $\{x_1, ..., x_t\}$ who received high attention from every position until $t$. Then, we record $S_{t+1 \to l} \in \{x_1, ...x_t\}$, tokens in $\{x_1, ..., x_t\}$ who received high attention from position after $t$. The persistence ratio is the intersection divided by the size of $S_{t+1 \to l}$. Formally,

$$Persistence\ Ratio = \frac{|S_{t+1 \to l} \cap S_{0 \to t}|}{|\{x | x \in S_{t+1 \to l}, x \in \{x_1, ..., x_t\}\}|}$$

At the same time, we measure $\frac{|S_{0 \to t}|}{t}$. $|S_{0 \to t}| = t$ indicates that every token substantially impacted at least one position, which is the trivial case of *persistence of importance hypothesis*. Our test is performed with OPT models [24] with different datasets such as OpenBookQA [25] and Wiki-Text [26]. In our verification, we set $t = \frac{l}{2}$, which measures the overlapping between the first and later half of the sentences. Same as in Section 3.1, we set $\alpha = \frac{1}{t}$, which suggests an average score.

**Result.** We present our main results in Figure 2. First, given the current criterion of pivotal token and $t$ value, the size of $S_{0 \to t}$ is considerably smaller than half of the sentence length. This verifies that we are not considering the trivial case of our hypothesis. Second, the persistence ratio is generally over 95%, with dips in the later transformer layers. The pivotal token set of the later half sentences is mostly included in the set of the first half sentences. Combining these two pieces of empirical evidence, we see positive evidence for our hypothesis test.

**Implication.** The hypothesis provides insights for understanding the behavior of LLMs and opens up new opportunities for reducing the KV cache memory. The hypothesis suggests the possibility of predicting the potentially influential tokens for future steps. The non-influential tokens are unnecessary to store in the memory, as they are unlikely to have high attention scores. This reduces the number of tokens stored in the KV cache and the computation required at the attention.

### 3.3 Attention Weights Decides the Pivotal Tokens

In the previous section, we verified that the significant tokens would continue to be significant. In this section, we try to understand the reasons for such phenomena. We consider the token generation process of a simplified model: a single-layer transformer model with single-head attention.

$$x_{t+1} = \mathcal{F}(a_t), \text{ where } a_t = \texttt{softmax}\left(1/t \cdot x_t W_Q W_K^\top X_{t-1}^\top\right) X_{t-1} W_V W_O \quad (1)$$

$x_t \in \mathbb{R}^{1 \times d}$ is a row vector. $X_{t-1} \in \mathbb{R}^{(t-1) \times d}$ denotes the aggregation of $x_1, ..., x_{t-1}$, where the $j$th row is $x_j$. $W_Q, W_K, W_V \in \mathbb{R}^{d \times d}$ and $W_O \in \mathbb{R}^{d \times d}$ are the attention weights. Lastly, $\mathcal{F} : \mathbb{R}^{1 \times d} \to \mathbb{R}^{1 \times d}$ denotes the MLP block following attention block, a two-layer MLP with skip connections, given by

$$\mathcal{F}(x) = x + W_2 \texttt{relu}(W_1 x) \quad (2)$$

---

**Algorithm 1** Inference with Budget KV cache

---

**Input**: Memory Budget $B$, Maximum Sequence Length $T_{\max}$
**Key Cache** $\bar{\mathcal{K}} \in R^{n \times d}$, **Value Cache** $\bar{\mathcal{V}} \in R^{n \times d}$, where $n = 0$
**while** $t < T_{\max}$ **do**
    Model update $\bar{\mathcal{K}}, \bar{\mathcal{V}}$ such that $n \leftarrow n + 1$
    **if** $n > B$ **then**:
        Compress KV cache using Algorithm 2 such that $n \leq B$.
    **end if**
    $t \leftarrow t + 1$
**end while**

---

---

**Algorithm 2** Compress KV Cache

---

**Input**: Key Cache $\bar{\mathcal{K}} \in \mathbf{R}^{n \times d}$, Value Cache $\bar{\mathcal{V}} \in R^{n \times d}$, History Window Size $w$, Recent Window Size $r$, Drop Amount $m$, Generation Step $t$,
Importance Record $I \leftarrow \vec{0} \in R^t$
**for** $i \in [t - w, t]$ **do**                          $\triangleright$ Consider tokens within history window
    $I \leftarrow I + \alpha_i < \frac{1}{t}$                  $\triangleright$ Increment the counter for low score token
**end for**
$I[: -r] \leftarrow 0$                             $\triangleright$ Keep cache within the recent window
Keep set $S_t \leftarrow Argsort\,(I)\,[: -m]$
Keep everything in $S_t$ in $\bar{\mathcal{K}} \in R^{n \times d}$, $\bar{\mathcal{V}} \in R^{n \times d}$ such that $n \leftarrow n - m$

---

We are interested in the attention scores $\alpha_t = \texttt{softmax}(1/t \cdot x_t W_Q W_K^\top X_{t-1}^\top)$. Notice that $\alpha_{t,j}$ scales with $x_t W_Q W_K^\top x_j^\top$. The following theorem characterizes the behavior of $x_t W_Q W_K^\top x_j^\top$

**Theorem 3.1.** *Let $A = W_V W_O W_Q W_K^\top$ and let $\lambda_K, \lambda_Q, \lambda_V, \lambda_O$ denote the largest singular values of $W_K, W_Q, W_V, W_O$, respectively. Consider the transformer in (1) with normalized inputs $\|x_t\|_2 = 1$ for all $t$. Let $c, \epsilon > 0$ be constants. Assume that $a_t x_{t+1}^\top \geq (1 - \delta) \|a_t\|_2$ with $\delta \leq \left( \frac{c\epsilon}{\lambda_Q \lambda_K \lambda_V \lambda_O} \right)^2$. Then for all $x_\ell$ satisfying $x_\ell A x_\ell^\top \geq c$ and $x_\ell A x_\ell \geq \epsilon^{-1} \max_{j \in [t], j \neq \ell} x_j A x_\ell^\top$, it holds that*

$$\frac{x_\ell A x_\ell^\top}{\|a_t\|_2}(\alpha_{t,\ell} - 3\epsilon) \leq x_{t+1} W_Q W_K^\top x_j^\top \leq \frac{x_\ell A x_\ell^\top}{\|a_t\|_2}(\alpha_{t,\ell} + 3\epsilon) \tag{3}$$

The proof is provided in Appendix B. Theorem 3.1 shows that under an assumption on the MLP in (2), for all $x_\ell$ such that $x_\ell A x_\ell^\top$ is large enough, $x_{t+1} W_Q W_K^\top x_j^\top$ satisfies Equation (3). The assumption on the MLP $a_t x_{t+1}^\top \geq (1 - \delta) \|a_t\|_2$ essentially requires a large cosine similarity between the input and output of $\mathcal{F}$. This behavior can be empirically verified in Appendix A. Essentially, skip connection dominates the output because $\|x\|_2 \gg \|W_2 \texttt{relu}(W_1 x)\|_2$, resulting in a cosine similarity close to one between input and output. Equation (3) shows that despite a factor of $\frac{x_\ell A x_\ell^\top}{\|a_t\|_2}$, $x_{t+1} W_Q W_K^\top x_j^\top$ almost scales with $\alpha_{t,\ell}$. Since $x_{t+1} W_Q W_K^\top x_j^\top$ directly affects $\alpha_{t+1,\ell}$, this property shows that a larger $\alpha_{t,\ell}$ will potentially imply a large $\alpha_{t+1,\ell}$.

Our theorem shows that the property in Equation (3) property only holds for $x_\ell$ such that $x_\ell A x_\ell^\top$ is large. $A$ are trained attention weights. This condition may suggest that the trained weights $A$ selects $x_\ell$ as a pivotal token. Each attention is learned to identify some subspace. Only those tokens embedded inside these regions are pivotal for this attention. This would explain why only some specific tokens are always relevant.

## 4 Sequential Token Generation Under budget

In this section, we present SCISSORHANDS, which reduces the KV cache memory from the sequence length dimension without fine-tuning the model. In Section 4.1, we describe how SCISSORHANDS maintains the KV cache under a given budget. Section 4.2 provides a theoretical analysis of the algorithm and the approximation error.

## 4.1 Budget KV Cache for Single Attention Head

In this section, for the sake of the discussion, we drop the layer number notation $i$ and batch size dimension. $\mathcal{K}_t, \mathcal{V}_t \in R^{t \times d}$ denote for the KV cache until step $t$. $x_t \in \mathbb{R}^{1 \times d}$ is a row vector that denotes the input to attention at step $t$. The output of an attention head at step $t$ can be written as,

$$a_t = \sum_{i=1}^{t} \alpha_{t,i} \mathcal{V}[i]_t, \text{ where } \alpha_{t,i} = \frac{\exp(\langle x_t W_Q, \mathcal{K}_t[i] \rangle)}{\sum_{i=1}^{t} \exp(\langle x_t W_Q, \mathcal{K}_t[i] \rangle)}$$

**Intuition.** As shown in Section 3, the attention scores $\alpha_{t,i}$ follow a strong power-law distribution. For the autoregressive generation process, if there exists an oracle such that we can identify the heavy score tokens before the future generation step, then the memory of the KV cache can be significantly reduced by only storing the heavy score tokens. Fortunately, the *persistence of importance hypothesis* provides us with such an oracle. It states that only historical tokens with significant contributions toward previous generated tokens will have significant contributions toward future tokens.

**Challenges.** LLMs are deployed on hardware with a fixed memory. The algorithm should maintain the cache under fixed memory to meet the hard requirement. Further, LLMs are already computationally intensive. The algorithm should avoid introducing much extra burden on computation.

A fixed memory budget for one attention head is $B$ tokens. In other words, we can store key and value embedding for $B$ previous tokens. We describe the problem as follows,

**Definition 4.1** (Sequential generation at an attention head under budget $B$). *Given a stream of token embedding, including prompt and previously generated tokens, denotes their input to the head as* $\{x_1, \ldots, x_t, \ldots\}$. *The problem of sequential generation at an attention head under budget $B$ is maintaining a key cache $\bar{\mathcal{K}}_t$ and value cache $\bar{\mathcal{V}}_t$ such that $\bar{\mathcal{K}}_t, \bar{\mathcal{V}}_t \in R^{n \times d}$ and $n < B$.*

**Approach.** Inspired by the textbook solution of reservoir sampling and the Least Recent Usage cache replacement algorithm, SCISSORHANDS reserves a fixed memory buffer for the KV cache. When the buffer is full, SCISSORHANDS drops stored but non-influential tokens from the cache. We present the main algorithm in Algorithm 1 and Algorithm 2.

When the KV cache size exceeds the budget, SCISSORHANDS drops tokens from the KV cache according to Algorithm 2. The importance record is a counter that indicates how many times a token is deemed non-important. We choose attention scores as the importance indicators, following our methodology in Section 3.2. The importance record is collected over a history window $w$ to reduce variance. A higher counter suggests dropping from the cache. Recent tokens are always kept because of the lack of information on their importance by setting the counter for all tokens in the recent window $r$ to 0.

With a sampled KV cache, attention output can be computed by the following estimator

$$\hat{a}_t = \sum_{i=1}^{n} \hat{\alpha}_{t,i} \bar{\mathcal{V}}_t[i], \text{ where } \hat{\alpha}_{t,i} = \frac{\exp(\langle x_t W_Q, \bar{\mathcal{K}}_t[i] \rangle)}{\sum_{i=1}^{n} \exp(\langle x_t W_Q, \bar{\mathcal{K}}_t[i] \rangle)}$$

**Overhead Tradeoff** At the compression step, an extra attention computation is introduced to collect the importance measurements over a history window. However, such compression is not required at every generation step. $m$ controls the frequency, and we use $m = 0.5B$ in our experiment. Further, steps after the compression have reduced attention computation because of the reduction in the KV cache. On the other hand, one can trade a tiny amount of memory to avoid the overhead by maintaining the importance record during generation steps in Algorithm 1.

**Allocating Budgets Across Attention Heads.** An LLM typically consists of $L$ transformer layers where each layer has $H$ heads. A total memory budget has to be distributed over layers and heads. Within each transformer layer, the budget is distributed evenly across heads. Within the entire model, we distributed the budget according to Figure 2. The rule of thumb is to allocate more budget to later layers to compensate for the lower persistence ratio.

## 4.2 Theoretical Analysis.

We study how much the tokens generated by the compressed KV cache deviate from the tokens generated by the original transformer using our simplified model in (1). Let $\{\tilde{x}_t\}_{t=0}^{T}$ denote the tokens generated by the transformer with budget KV cache as in Algorithm 2 with $m = 1$:

$$\tilde{x}_{t+1} = \mathcal{F}\left(\tilde{a}_t\right), \text{ where } \tilde{a}_t = \texttt{softmax}\left(1/t \cdot \tilde{x}_t W_Q \tilde{\mathcal{K}}_t^{\top}\right) \tilde{\mathcal{V}}_t^{\top} W_O$$

Notice that when $m = 1$, i.e., in each iteration, we drop one token with the lowest score, the cache will always maintain $B$ tokens. If the ranking of the attention scores does not change in each iteration, Algorithm 2 will always drop tokens with the smallest attention scores.

For reference purposes, let $\{x_t\}_{t=0}^{T}$ denote the tokens generated by a vanilla transformer defined in (1). We will bound the difference $\|x_t - \tilde{x}_t\|_2$.

**Theorem 4.1.** *Let $\lambda_1, \lambda_2$ denote the largest singular values of $W_1$ and $W_2$ in (2). Let*

$$\beta_{t,j} = \frac{\exp\left(1/t \cdot \tilde{x}_t W_Q W_K^{\top} \tilde{x}_j^{\top}\right)}{\sum_{i=1}^{t-1} \exp\left(1/t \cdot \tilde{x}_t W_Q W_K^{\top} \tilde{x}_i^{\top}\right)}$$

*and assume that each $\beta_{t,j} = cv_{t,j}$, where $v_{t,j}$ are sampled from a power-law distribution with pdf $f(x) = c(x + b)^{-k}$. Suppose that $\lambda_V \lambda_O (1 + \lambda_1 \lambda_2)(1 + \lambda_Q \lambda_K) \leq \frac{1}{2}$. Let $T_{\min}$ and $T_{\max}$ denote the starting and maximum sequence lengths, respectively, and let $B \leq T_{\max}$ denote the budget as in Algorithm 2. If for all $t \in [T_{\min}, T_{\max}]$, $S_t$ contains only tokens with at most the largest $B$ values of $\beta_{t,j}$, that is, $|S_t| = B$ and $\min_{j \in S_t} \beta_{t,j} \geq \max_{j \notin \hat{S}_t} \beta_{t,j}$, then for all $\epsilon \in (0,1)$, with probability at least $1 - T_{\max} \exp\left(-\frac{\epsilon^2 b^2 (T_{\min}-1)}{(k-2)^2(u-b)^2}\right) - T_{\max} \exp\left(-\frac{2(T_{\min}-1)(1-B/T_{\max})^2}{(1-\epsilon)^2}\right)$, the following error bound must hold for all $t \in [T_{\min}, T_{\max}]$*

$$\mathbb{E}\left[\|x_t - \tilde{x}_t\|_2\right] \leq \frac{2.1(1 - B/T_{\max})}{(1-\epsilon)^2}\left(k - (k-1)\left(\frac{1-\epsilon}{B/T_{\max} - \epsilon}\right)^{1/(k-1)}\right) \tag{4}$$

The definition of $\beta_{t,j}$ means the attention scores computed on the tokens generated by the compressed approach. Our theorem assumes that dropping the tokens depends on the attention score of the current iteration. (4) provided a bound on the expected difference between the tokens generated in the budget and the original approach. The upper bound scales with $1 - B/T_{\max}$. When $B = T_{\max}$, meaning that we are keeping all of the tokens, the error becomes zero. The term $k - (k-1)\left(\frac{1-\epsilon}{B-\epsilon}\right)^{1/(k-1)}$ depends on the distribution that the attention scores are fitted to and is always less than one. With a strong power-law distribution, this term provides a further decrease to the error bound in (4).

## 5 Empirical Evaluation

In this section, we present the results that demonstrate SCISSORHANDS achieves up to $5\times$ reduction in the KV cache memory compared to the standard model with no accuracy loss. We also show that SCISSORHANDS is compatible with 4-bit quantization.

**Experiment Setting.** We compare the accuracy of SCISSORHANDS-OPT against the original OPT on one language model datasets C4 [23] and a number of few-shot downstream tasks: Hellaswag [27], MathQA [28], PIQA [29], Winogrande [30]. We use lm-eval-harness [31] to evaluate few-shot tasks. Our experiments are conducted on NVIDIA 4 A100 40GB GPU servers.

**No Accuracy Drop untill $5\times$.** In Figure 3, we present SCISSORHANDS's accuracy trend where $1\times$ denotes the original OPT. In the language modeling setting, perplexity is the lower the better. For OPT-6B, perplexity is maintained until 50% of the original KV cache size for OPT-13B. For OPT-66B, perplexity is maintained until 75% of the original KV cache. We observe a flatter accuracy trend as the model size grows, which is exceptionally encouraging. This suggests that SCISSORHANDS can scale with the model size. Downstream tasks are usually less sensitive to perturbation and bear more variance in terms of accuracy. We evaluate the 5-shot setting and $1\times$ denotes the original OPT model. For Winogrande and MathQA, accuracy is maintained even after $5\times$ compression for OPT-66B. Similar to the language modeling setting, SCISSORHANDS performs better at larger models. Generally, accuracy is maintained with 15% - 30% of the original KV cache size.

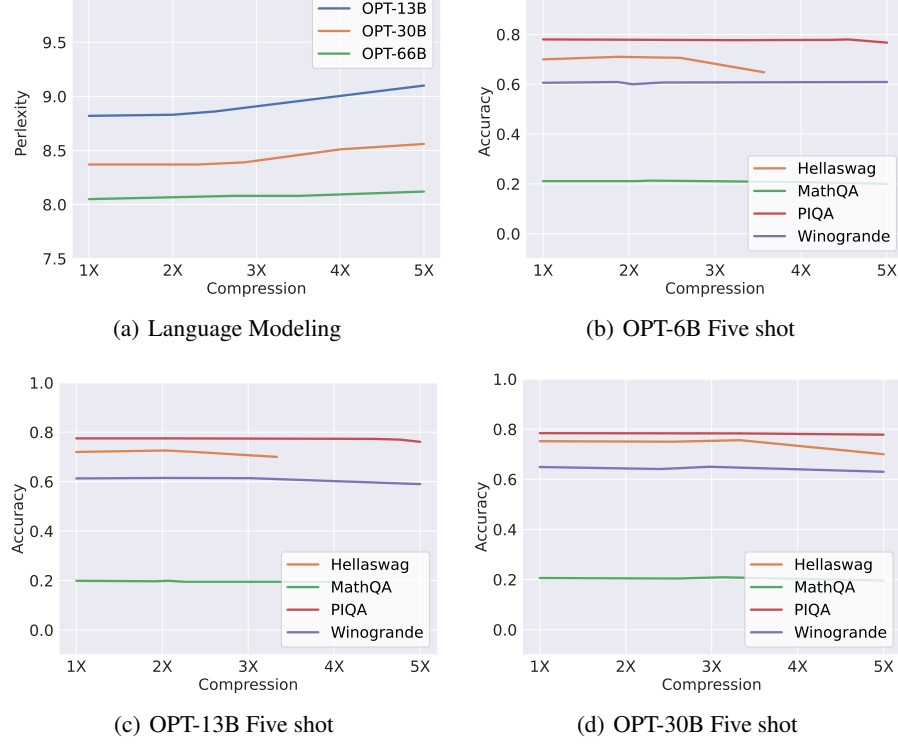

| (a) Language Modeling | (b) OPT-6B Five shot |
|---|---|
| (c) OPT-13B Five shot | (d) OPT-30B Five shot |

Figure 3: Accuracy trend of SCISSORHANDS on language modeling dataset and downstream tasks with different KV cache compression. In general, SCISSORHANDS incurs no accuracy drop until $5\times$ compression on OPT-66B.

**Ablation on the Importance of Pivotal Tokens** We divide C4 into three subsets depending on the sequence length. C4-[256-512] contains data sequences that are longer than 256 tokens but less than 512 tokens. C4-[512-1024] contains data sequences longer than 512 tokens but less than 1024 tokens. C4-[1024-2048] contains data sequences that are longer than 1024 tokens but less than 2048. Results are summarized in Table 3. Local Windows refers to only keeping tokens in the recent window, while SCISSORHANDSkeeps both recent tokens and pivotal tokens. We observe the perplexity of the full model degrades slightly with the growing sequence length. At all sequence lengths, SCISSORHANDS's performance is comparable against the full cache model, while Local Window incurs a significant quality loss. This demonstrates that keeping the pivotal tokens is important to reserve model performance. It is also interesting to note that at longer sequence lengths, the local window has higher accuracy. This also shows at longer sequence length, the attention mechanism in current architecture tends to focus on recent context.

Table 3: Perplexity on C4 with different sequence lengths.

|  | [256 - 512] | [512- 1024] | [1024- 2048] |
|---|---|---|---|
| OPT-13B | 8.7968 | 9.1017 | 9.3005 |
| OPT-13B + Local Window | 81.8297 | 29.3823 | 15.5883 |
| OPT-13B + SCISSORHANDS | 8.7972 | 9.1011 | 9.3009 |

Table 4: Applying 4-bit quantization on top of SCISSORHANDS on Hellaswag.

| OPT-6B | | |
|---|---|---|
| Original | SCISSORHANDS | SCISSORHANDS+ 4-bit |
| 0.702 | 0.706 | 0.704 |
| OPT-13B | | |
| Original | SCISSORHANDS | SCISSORHANDS+ 4-bit |
| 0.720 | 0.720 | 0.720 |

**Compatible with 4-bit Quantization** We test the compatibility of quantization and SCISSORHANDS at $2\times$ compression. We adopt 4-bit quantization following [7]. Even Hellaswag is most sensitive based on Figure 3, adding quantization doesn't introduce compounded errors.

**Ablation on Attention Score Error.** We present the change ratio in attention score between original OPT-13B and SCISSORHANDS OPT-13B at $3\times$ compression on C4 in Figure 4.

We observe the attention score generated from SCISSORHANDS is almost the same as the original KV cache, which also echoes Theorem 4.1. The change ratio is calculated as $\frac{\alpha_s - \alpha_o}{\alpha_o}$ where $\alpha_s$ is the SCISSORHANDS attention score and $\alpha_o$ is the original score. From Figure 4, we observe that the change ratio is centered around 0. -1 indicating that $\alpha_s$ is significantly smaller compared to the original, suggesting that a small portion of the important tokens are dropped in the cache. To explain the above observation of SCISSORHANDS, we denote the $n$ number of tokens with the highest score as $\{x_t^{top\_n}\}_{t=0}^T$. Then, for any other sets of tokens $\{x_t'\}_{t=0}^T$ that has no greater than $n$ tokens, we can easily prove that $similarity\left(x_t^{topB}, x_t\right) \leq (x_t', x_t)$. Thus, SCISSORHANDS gives the most similar output as the original model at all layers.

## 6 Discussion, Limitation, and Future Work

We discover repetitive attention patterns given trained language models. One interesting question that needs to be answered is whether such behavior is a model architecture bias or an unexpected training outcome. For such purpose, we perform the same experiment with a randomly initialized OPT, and compare it against the results presented in Section 3.1. As shown in Figure 5, the repetitive attention pattern does not exist in randomly initialized models. Apart from an efficiency deployment perspective, could such repetitive attention patterns contribute to some known problems in language generation, such as repetitions? It may be worth investigating the relationship between repetitive attention patterns and undesired generations.

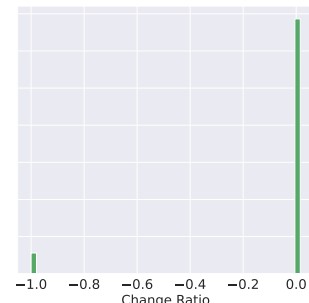

Figure 4: Score between OPT and SCISSORHANDS.

Due to the limitation of the server in academics, the largest model we can fit is OPT-66B. We try to understand the behavior and verify the generality across the different models and datasets. However, we cannot access the training process and fail to know exactly how an LLM is trained to exhibit such behavior. Experiments with the large model create carbon dioxide emissions. However, our work improves the efficiency of LLM, and we foresee no negative impacts.

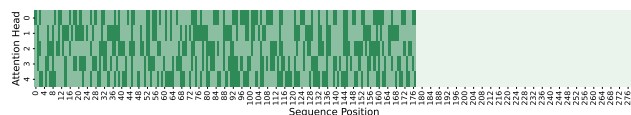

(a) Attention map of the token at position 178

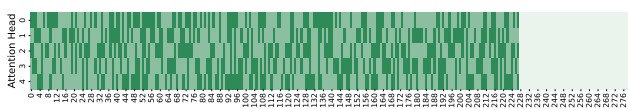

(b) Attention map of the token at position 228

Figure 5: We plot the attention map corresponding to Section 3.1 but with a randomly initialized OPT. We observe no repetitive attention for a randomly initialized model.

## 7 Conclusion

Inspired by our intriguing findings that pivotal tokens exert a lasting influence on future steps, we developed SCISSORHANDS to leverage this observation to reduce the memory usage of KV cache. Our method achieves memory reductions of $5\times$ in the KV cache without compromising the performance of LLMs. Furthermore, we demonstrate the compatibility of SCISSORHANDS with quantization techniques, opening up the possibility of reducing memory usage in both the representation and sequence length dimensions.

## 8 Acknowledgement

We would like to thank the anonymous reviewers for their helpful discussions and feedback. This work is supported by NSF-CCS-2211815, ONR-DURIP and NSF-BIGDATA-1838177.

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

# Appendix

## A    More Results

### A.1    Repetitive Attention Pattern

We provide the attention map similar to Figure 1 but from a different transformer layer on the same text in Figure 6, Figure 7, Figure 8 and Figure 9. A repetitive pattern and attention sparsity can be observed across layers.

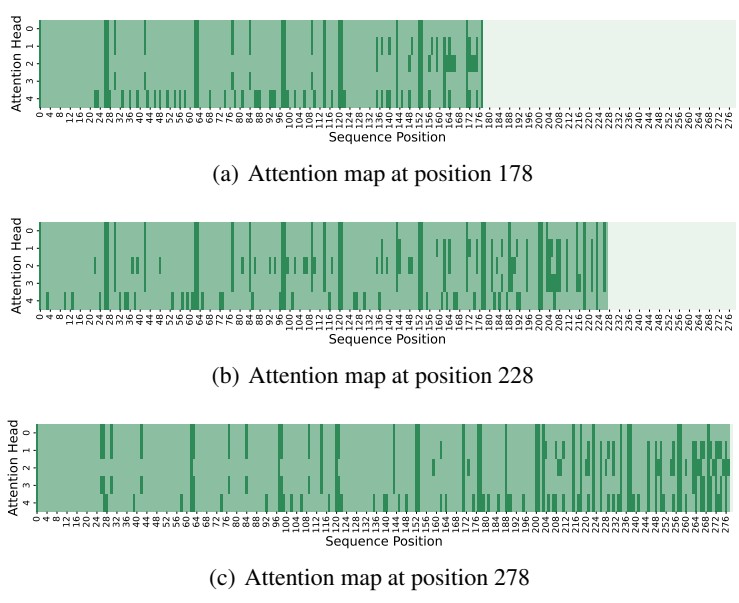

(a) Attention map at position 178

(b) Attention map at position 228

(c) Attention map at position 278

Figure 6:   Attention Map at Layer 5

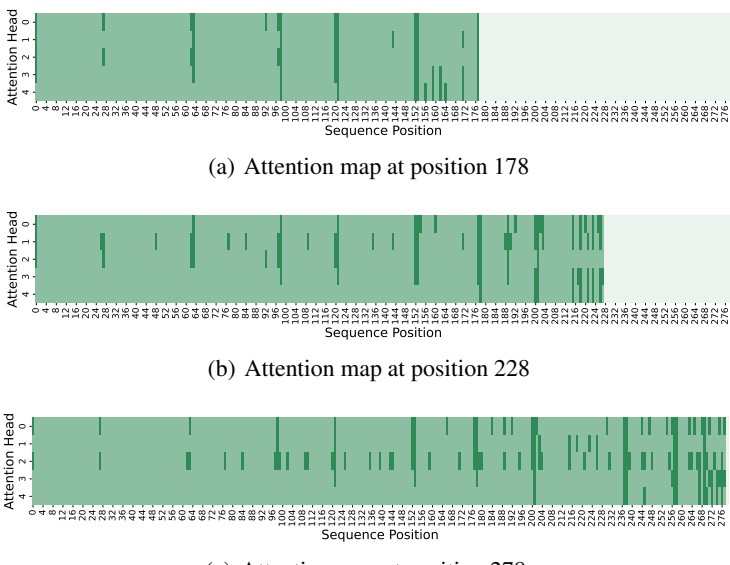

(a) Attention map at position 178

(b) Attention map at position 228

(c) Attention map at position 278

Figure 7:   Attention Map at Layer 10

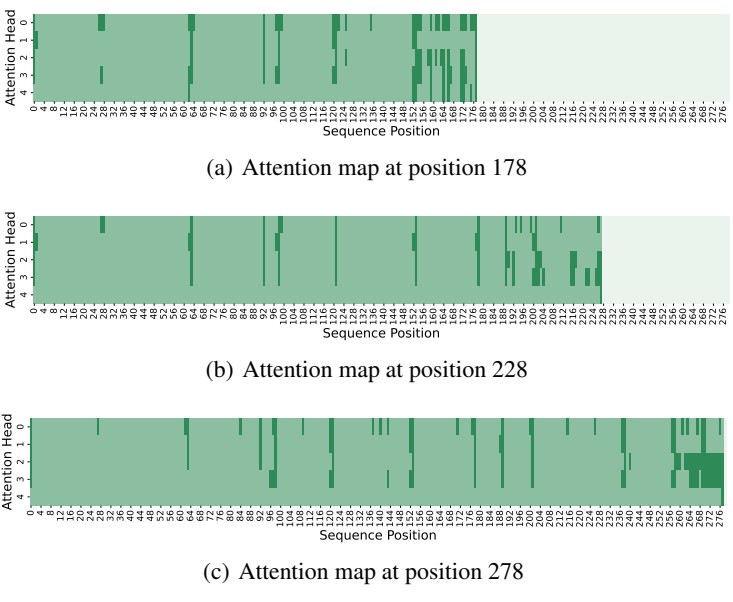

(a) Attention map at position 178

(b) Attention map at position 228

(c) Attention map at position 278

Figure 8: Attention Map at Layer 15

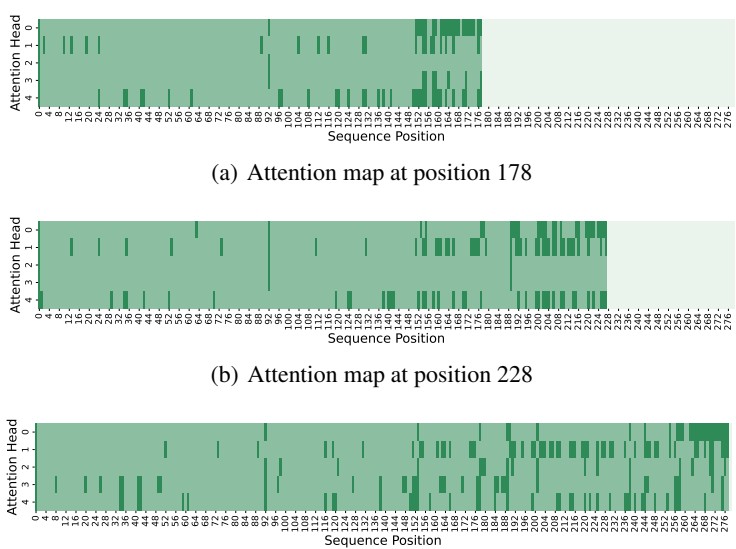

(a) Attention map at position 178

(b) Attention map at position 228

(c) Attention map at position 278

Figure 9: Attention Map at Layer 20

## A.2 Cross Layer Cosine Similarity

In Section 3.3, our analysis assumes a large cosine similarity between the input and output of $\mathcal{F}$. Here, we provide empirical evidence to support such an assumption in Figure 10. Because of the residual connection in $\mathcal{F}$ and the domination of $x$, the cosine similarity between $x$ and $\mathcal{F}(x)$ is extremely high.

## A.3 Generated examples with SCISSORHANDS

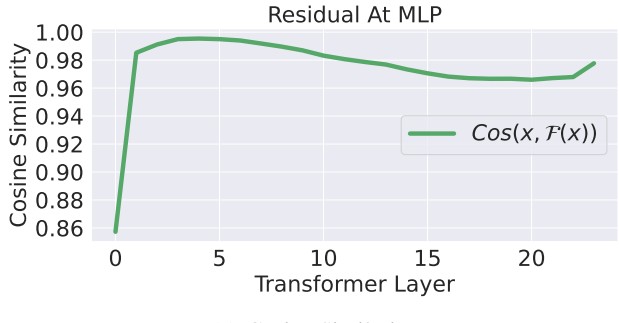

(a) Cosine Similarity

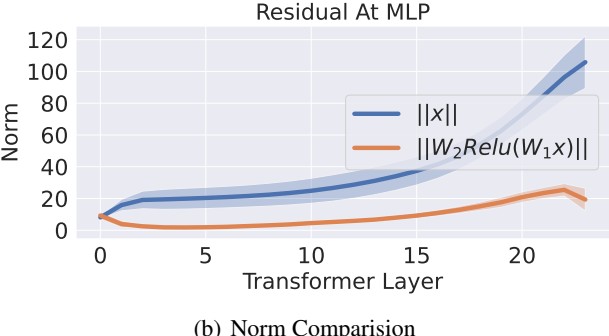

(b) Norm Comparision

Figure 10: $x$ and $\mathcal{F}(x)$ is high in cosine similarity

Table 5: Generated examples using OPT-13B with full cache and SCISSORHANDSat different compression ratio.

| | |
|---|---|
| Prompt | Hi. Does Asus ever plan to update the intel and nvidia display drivers on the UL30VT laptop? The only downloadable drivers from the Asus website are at least 3 years behind and new features for both graphics chips have been released since then. I sent a message to Asus but no response was given. Intel and Nvidia will not allow the user to download the new drivers directly. problem is the drivers won't install. states something about your notebook needs custom drivers from the manufacturer due to customized this that or the other. |
| OPT-13B | I have the same problem. I have a laptop with an Intel HD Graphics 4000 and an Nvidia GeForce GT 740M. I can't install the drivers from the Asus website. I have to use the drivers from the Nvidia website. |
| 3x Compression | I have the same problem. I have a laptop with an Intel HD Graphics 4000 and an Nvidia GeForce GT 740M. I can't install the drivers from the Asus website. I have to use the drivers from the Nvidia website. |
| 6x Compression | I have the same problem. I have a custom laptop and I can't install the drivers. |

# B  Proofs

## B.1  Proof of Theorem 3.1

We consider the token generation process of a simplified model: a single-layer transformer model with single-head attention.

$$x_{t+1} = \mathcal{F}(a_t), \text{ where } a_t = \texttt{softmax}\left(1/t \cdot x_t W_Q W_K^\top X_{t-1}^\top\right) X_{t-1} W_V W_O \quad (5)$$

$x_t \in \mathbb{R}^{1 \times d}$ is a row vector. $X_{t-1} \in \mathbb{R}^{(t-1) \times d}$ denotes the aggregation of $x_1, \ldots, x_{t-1}$, where the $j$th row is $x_j$. $W_Q, W_K, W_V \in \mathbb{R}^{d \times p}$ and $W_O \in \mathbb{R}^{p \times d}$ are the attention weights. Lastly, $\mathcal{F} : \mathbb{R}^{1 \times d} \to \mathbb{R}^{1 \times d}$ denotes the MLP block following attention block, a two-layer MLP with skip connections, given by

$$\mathcal{F}(x) = x + W_2 \texttt{relu}(W_1 x) \quad (6)$$

We are interested in the attention scores $\alpha_t = \texttt{softmax}(1/t \cdot x_t W_Q W_K^\top X_{t-1}^\top)$. Notice that $\alpha_{t,j}$ scales with $x_t W_Q W_K^\top x_j^\top$. We first re-state the Theorem 3.1 below.

**Theorem B.1.** *Let $A = W_V W_O W_Q W_K^\top$ and let $\lambda_K, \lambda_Q, \lambda_V, \lambda_O$ denote the largest singular values of $W_K, W_Q, W_V, W_O$, respectively. Consider the transformer in (5) with normalized inputs $\|x_t\|_2 = 1$ for all $t$. Let $c, \epsilon > 0$ be constants. Assume that $a_t x_{t+1}^\top \geq (1 - \delta) \|a_t\|_2$ with $\delta \leq \left( \frac{c\epsilon}{\lambda_Q \lambda_K \lambda_V \lambda_O} \right)^2$. Then for all $x_\ell$ satisfying $x_\ell A x_\ell^\top \geq c$ and $x_\ell A x_\ell \geq \epsilon^{-1} \max_{j \in [t], j \neq \ell} x_j A x_\ell^\top$, it holds that*

$$\frac{x_\ell A x_\ell^\top}{\|a_t\|_2} (\alpha_{t,\ell} - 3\epsilon) \leq x_{t+1} W_Q W_K^\top x_j^\top \leq \frac{x_\ell A x_\ell^\top}{\|a_t\|_2} (\alpha_{t,\ell} + 3\epsilon) \tag{7}$$

As a preparation of the proof, we first show two lemmas.

**Lemma B.1.** *Let $x_1, x_2 \in \mathbb{R}^{1 \times m}$ satisfies $\|x_1\|_2 = \|x_2\|_2 = 1$ and $x_1 x_2^\top \geq 1 - \delta$ for some $\delta \in (0, 1)$. Then for all $y \in \mathbb{R}^{1 \times m}$ we have*

$$\left| x_1 y^\top - x_2 y^\top \right| \leq \sqrt{2\delta} \|y\|_2$$

*Proof.* Let $x_2 = x_2^\| + x_2^\perp$ where

$$x_2^\| = x_1 x_2^\top \cdot x_1; \quad x_2^\perp = x_2 - x_2^\|$$

Then it is easy to see that $x_2^\perp x_1^\top = 0$. By the Pythagorean Theorem, we have

$$\left\| x_2^\perp \right\|_2^2 = \|x_2\|_2^2 - \left\| x_2^\| \right\|_2^2 = \delta(2 - \delta)$$

Therefore, we have

$$\begin{aligned}
\|x_1 - x_2\|_2^2 &= \left\| (x_1 - x_2^\|) - x_2^\perp \right\|_2^2 \\
&= \left\| \left( 1 - x_1 x_2^\top \right) x_1 - x_2^\perp \right\|_2^2 \\
&= \left( 1 - x_1 x_2^\top \right)^2 + \left\| x_2^\perp \right\|_2^2 \\
&= 2\delta
\end{aligned}$$

Thus, the Cauchy-Schwarz inequality implies

$$\left| x_1 y^\top - x_2 y^\top \right| \leq \|x_1 - x_2\|_2 \cdot \|y\|_2 = \sqrt{2\delta} \|y\|_2$$

$\square$

**Lemma B.2.** *Let $\ell \in [t]$ be given. Suppose that $x_\ell A x_\ell^\top > \epsilon^{-1} \left| x_j A x_\ell^\top \right|$ for all $j \neq \ell$. Then we have*

$$(\mathcal{S}(t)_\ell - \epsilon) x_\ell^\top a x_\ell \leq x_\ell^\top W_K^\top W_Q a_t \leq (\mathcal{S}(t)_\ell + \epsilon) x_\ell^\top a x_\ell$$

*Proof.* Notice that

$$a_t = \alpha_t X_{t-1} W_V W_O = \left( \sum_{j=1}^{t-1} \alpha_{t,j} x_j \right) W_V W_O$$

Thus, we have

$$a_t W_Q W_K^\top x_\ell^\top = \left( \sum_{j=1}^{t-1} \alpha_{t,j} x_j \right) W_V W_O W_Q W_K^\top x_\ell^\top = \sum_{j=1}^{t-1} \alpha_{t,j} x_j A x_\ell^\top$$

Therefore

$$\begin{aligned}
\left| a_t W_Q W_K^\top x_\ell^\top - \alpha_{t,\ell} x_\ell A x_\ell^\top \right| &= \left| \sum_{j=1, j \neq \ell}^{t-1} \alpha_{t,j} x_j A x_\ell^\top \right| \\
&\leq \sum_{j=1, j \neq \ell}^{t-1} \alpha_{t,j} \left| x_j A x_\ell^\top \right| \\
&\leq \epsilon x_\ell A x_\ell^\top \sum_{j=1, j \neq \ell}^{t-1} \alpha_{t,j} \\
&\leq \epsilon x_\ell A x_\ell^\top
\end{aligned}$$

where in the second inequality we use $\epsilon^{-1} \left| x_j A x_\ell^\top \right| \leq x_\ell A x_\ell^\top$ and in the third inequality we use $\sum_{j=1, j \neq \ell}^{t-1} \alpha_{t,j} \leq \sum_{j=1}^{t-1} \alpha_{t,j} = 1$. This implies that

$$(\alpha_{t,\ell} - \epsilon) x_\ell A x_\ell^\top \leq a_t W_Q W_K^\top x_\ell^\top \leq (\alpha_{t,\ell} + \epsilon) x_\ell A x_\ell^\top$$

$\square$

Now we proceed to the main body of the proof. Assume that $\|x_\ell\|_2 = 1$ for all $\ell$. Using Lemma (B.1), if $a_t x_{t+1}^\top \geq (1 - \delta) \|a_t\|_2$, then we have

$$\left| \|a_t\|_2^{-1} a_t W_Q W_K^\top x_\ell^\top - x_{t+1} W_Q W_K^\top x_\ell^\top \right| \leq \sqrt{2\delta} \left\| W_Q W_K^\top x_\ell^\top \right\|_2$$

Recall that $\lambda_Q, \lambda_K$ are the maximum singular values of $W_Q$ and $W_K$, respectively. Then it holds that $\left\| W_Q W_K^\top x_\ell^\top \right\|_2 \leq \lambda_Q \lambda_K \|x_\ell\|_2$. Using $\|x_\ell\|_2 = 1$, we have

$$\left| \|a_t\|_2^{-1} a_t W_Q W_K^\top x_\ell^\top - x_{t+1} W_Q W_K^\top x_\ell^\top \right| \leq \sqrt{2\delta} \lambda_Q \lambda_K$$

Notice that

$$\|a_t\|_2 = \left\| \left( \sum_{j=1}^{t-1} \alpha_{t,j} x_j \right) W_V W_O \right\|$$

$$\leq \lambda_O \lambda_V \left\| \sum_{j=1}^{t-1} \alpha_{t,j} x_j \right\|_2$$

$$\leq \lambda_O \lambda_V \sum_{j=1}^{t-1} \alpha_{t,j} \|x_j\|_2$$

$$= \lambda_O \lambda_V$$

Then since $\delta \leq \left( \frac{c\epsilon}{\lambda_Q \lambda_K \lambda_V \lambda_O} \right)^2$, we have

$$\left| \|a_t\|_2^{-1} a_t W_Q W_K^\top x_\ell^\top - x_{t+1} W_Q W_K^\top x_\ell^\top \right| \leq \frac{2c\epsilon}{\lambda_V \lambda_O} \leq \frac{2c\epsilon}{\|a_t\|_2}$$

Since by Lemma (B.2), we have

$$\left| a_t W_Q W_K^\top x_\ell^\top - \alpha_{t,\ell} x_\ell A x_\ell^\top \right| \leq \epsilon x_\ell^\top a x_\ell$$

It must hold that

$$\left| x_{t+1} W_Q W_K^\top x_\ell^\top - \|a_{t+1}\|_2^{-1} \alpha_{t,\ell} x_\ell A x_\ell^\top \right| \leq \frac{\epsilon}{\|a_t\|_2} x_\ell^\top a x_\ell + \frac{2c\epsilon}{\|a_t\|_2}$$

Since $x_\ell^\top a x_\ell \geq c$, it holds that

$$\frac{2c\epsilon}{\|a_t\|_2} \leq \frac{2\epsilon}{\|a_t\|_2} x_\ell^\top a x_\ell$$

which implies that

$$\left| x_{t+1} W_Q W_K^\top x_\ell^\top - \|a_t\|_2^{-1} \alpha_{t,\ell} x_\ell A x_\ell^\top \right| \leq \frac{3\epsilon}{\|a_t\|_2} x_\ell^\top a x_\ell$$

Therefore

$$\frac{x_\ell A x_\ell^\top}{\|a_t\|_2} (\alpha_{t,\ell} - 3\epsilon) \leq x_{t+1} W_Q W_K^\top x_\ell^\top \leq \frac{x_\ell A x_\ell^\top}{\|a_t\|_2} (\alpha_{t,\ell} + 3\epsilon)$$

## B.2 Proof of Theorem 4.1

Let $\{\tilde{x}_t\}_{t=0}^T$ denote the tokens generated by the transformer with budget KV cache as in Algorithm 2 with $m = 1$:

$$\tilde{x}_{t+1} = \mathcal{F}(\tilde{a}_t), \text{ where } \tilde{a}_t = \texttt{softmax}\left(1/t \cdot \tilde{x}_t W_Q \tilde{\mathcal{K}}_t^\top\right) \tilde{\mathcal{V}}_t^\top W_O$$

Notice that when $m = 1$, i.e., in each iteration, we drop one token with the lowest score, the cache will always maintain $B$ tokens. If the ranking of the attention scores does not change in each iteration, Algorithm 2 will always drop tokens with the smallest attention scores.

For reference purposes, let $\{x_t\}_{t=0}^T$ denote the tokens generated by a vanilla transformer defined in (5). We re-state Theorem 4.1 below, which bounds the difference $\|x_t - \tilde{x}_t\|_2$.

**Theorem B.2.** *Let* $\lambda_1, \lambda_2$ *denote the largest singular values of* $W_1$ *and* $W_2$ *in (6). Let*

$$\beta_{t,j} = \frac{\exp\left(1/t \cdot \tilde{x}_t W_Q W_K^\top \tilde{x}_j^\top\right)}{\sum_{i=1}^{t-1} \exp\left(1/t \cdot \tilde{x}_t W_Q W_K^\top \tilde{x}_i^\top\right)}$$

*and assume that each* $\beta_{t,j} = cv_{t,j}$*, where* $v_{t,j}$ *are sampled from a power-law distribution with pdf* $f(x) = c(x+b)^{-k}$*. Suppose that* $\lambda_V \lambda_O(1 + \lambda_1\lambda_2)(1 + \lambda_Q\lambda_K) \leq \frac{1}{2}$*. Let* $T_{\min}$ *and* $T_{\max}$ *denote the starting and maximum sequence lengths, respectively, and let* $B \leq T_{\max}$ *denote the budget as in Algorithm 2. If for all* $t \in [T_{\min}, T_{\max}]$*,* $S_t$ *contains only tokes with at most the largest* $B$ *values of* $\beta_{t,j}$*, that is,* $|S_t| = B$ *and* $\min_{j \in S_t} \beta_{t,j} \geq \max_{j \notin \hat{S}_t} \beta_{t,j}$*, then for all* $\epsilon \in (0,1)$*, with probability at least* $1 - T_{\max} \exp\left(-\frac{\epsilon^2 b^2 (T_{\min}-1)}{(k-2)^2(u-b)^2}\right) - T_{\max} \exp\left(-\frac{2(T_{\min}-1)(1-B/T_{\max})^2}{(1-\epsilon)^2}\right)$*, the following error bound must hold for all* $t \in [T_{\min}, T_{\max}]$

$$\mathbb{E}\left[\|x_t - \tilde{x}_t\|_2\right] \leq \frac{2.1(1 - B/T_{\max})}{(1-\epsilon)^2}\left(k - (k-1)\left(\frac{1-\epsilon}{B/T_{\max} - \epsilon}\right)^{1/(k-1)}\right)$$

Define $m_{k,j} = \mathbb{I}\{j \in S_t\}$. With the definition of $m_{k,j}$, $\tilde{a}_t$ can be written as

$$\tilde{a}_t = \left(\sum_{j=1}^{t-1} \tilde{\alpha}_{t,j} \tilde{x}_j\right) W_V W_O; \quad \tilde{\alpha}_{t,j} = \frac{m_{k,j} \exp\left(1/t \cdot \tilde{x}_t W_Q W_K^\top \tilde{x}_j^\top\right)}{\sum_{i=1}^{t-1} m_{k,j} \exp\left(1/t \cdot \tilde{x}_t W_Q W_K^\top \tilde{x}_i^\top\right)} \tag{8}$$

Our first lemma shows the Lipschitzness of the attention module.

**Lemma B.3.** *Consider two sequences of tokens* $\{x_i\}_{i=1}^t$ *and* $\{y_i\}_{i=1}^t$ *where* $\|x_i\|_2 = \|y_i\|_2 = 1$ *for all* $i \in [t]$*. Define* $X_{t-1}, Y_{t-1} \in \mathbb{R}^{(t-1)\times d}$ *as the matrices whose ith row are* $x_i$ *and* $y_i$*, respectively. Let* $\Delta_t = \|x_t - y_t\|_2$*. Then we have*

$$\left\|\texttt{softmax}\left(\frac{1}{t}x_t W_Q W_K^\top X_{t-1}^\top\right) - \texttt{softmax}\left(\frac{1}{t}y_t W_Q W_K^\top Y_{t-1}^\top\right)\right\|_2 \leq 2\frac{\sqrt{t-1}}{t}\lambda_Q\lambda_K\Delta_t$$

*Proof.* We can decompose the difference as

$$\left\|\texttt{softmax}\left(\frac{1}{t}x_t W_Q W_K^\top X_{t-1}^\top\right) - \texttt{softmax}\left(\frac{1}{t}y_t W_Q W_K^\top Y_{t-1}^\top\right)\right\|_2$$

$$\leq \left\|\texttt{softmax}\left(\frac{1}{t}x_t W_Q W_K^\top X_{t-1}^\top\right) - \texttt{softmax}\left(\frac{1}{t}x_t W_Q W_K^\top Y_{t-1}^\top\right)\right\|_2$$

$$+ \left\|\texttt{softmax}\left(\frac{1}{t}x_t W_Q W_K^\top Y_{t-1}^\top\right) - \texttt{softmax}\left(\frac{1}{t}y_t W_Q W_K^\top Y_{t-1}^\top\right)\right\|_2$$

By the Lipschitzness of softmax, we have

$$\left\|\texttt{softmax}\left(\frac{1}{t}x_t W_Q W_K^\top X_{t-1}^\top\right) - \texttt{softmax}\left(\frac{1}{t}x_t W_Q W_K^\top Y_{t-1}^\top\right)\right\|_2$$

$$\leq \frac{1}{t}\left\|x_t W_Q W_K^\top \left(X_{t-1} - Y_{t-1}\right)^\top\right\|_2$$

$$\leq \frac{1}{t}\lambda_Q\lambda_K \|x_t\|_2 \|X_{t-1} - Y_{t-1}\|_2$$

Since $\|x_t\|_2 = 1$ and $\|X_{t-1} - Y_{t-1}\|_2 = \left(\sum_{j=1}^{t-1} \|x_j - y_j\|_2\right)^{\frac{1}{2}} \le \sqrt{t-1}\Delta_t$, we have

$$\left\|\mathtt{softmax}\left(x_t W_Q W_K^\top X_{t-1}^\top\right) - \mathtt{softmax}\left(x_t W_Q W_K^\top Y_{t-1}^\top\right)\right\|_2 \le \frac{\sqrt{t-1}}{t}\lambda_Q\lambda_K\Delta_t$$

Similarly,

$$\left\|\mathtt{softmax}\left(\frac{1}{t}x_t W_Q W_K^\top Y_{t-1}^\top\right) - \mathtt{softmax}\left(\frac{1}{t}y_t W_Q W_K^\top Y_{t-1}^\top\right)\right\|_2$$

$$\le \frac{1}{t}\left\|(x_t - y_t)W_Q W_K^\top Y_{t-1}^\top\right\|_2$$

$$\le \frac{1}{t}\lambda_Q\lambda_K \|Y_{t-1}\|_F \|x_t - y_t\|_2$$

Since $\|x_t - y_t\|_2 = \Delta_t$ and $\|Y_{t-1}\|_2 = \sqrt{t-1}$, we have

$$\left\|\mathtt{softmax}\left(\frac{1}{t}x_t W_Q W_K^\top Y_{t-1}^\top\right) - \mathtt{softmax}\left(\frac{1}{t}y_t W_Q W_K^\top Y_{t-1}^\top\right)\right\|_2 \le \frac{\sqrt{t-1}}{t}\lambda_Q\lambda_K\Delta_t$$

Combining the two bounds gives

$$\left\|\mathtt{softmax}\left(\frac{1}{\sqrt{t}}x_t W_Q W_K^\top X_{t-1}^\top\right) - \mathtt{softmax}\left(\frac{1}{\sqrt{t}}y_t W_Q W_K^\top Y_{t-1}^\top\right)\right\|_2 \le 2\frac{\sqrt{t-1}}{t}\lambda_Q\lambda_K\Delta_t$$

$\square$

Our second lemma shows the difference between the output of the sampled and vanilla transformer when the input is the same.

**Lemma B.4.** *Let $\tilde{a}_t$ be defined as in (8). Define $b_t$ as*

$$b_t = \left(\sum_{j=1}^{t-1} \beta_{t,j}\tilde{x}_j\right)W_V W_O; \quad \beta_{t,j} = \frac{\exp\left(1/t \cdot \tilde{x}_t W_Q W_K^\top \tilde{x}_j^\top\right)}{\sum_{i=1}^{t-1}\exp\left(1/t \cdot \tilde{x}_t W_Q W_K^\top \tilde{x}_i^\top\right)} \tag{9}$$

*Assume that $\|x_j\|_2 = 1$ for all $j \in [t]$. Then we have*

$$\|\tilde{a}_t - b_t\|_2 \le \lambda_V\lambda_O \sum_{j\notin\hat{S}_t} \beta_{t,j}$$

*Proof.* A direction computation yields

$$\tilde{a}_t - b_t = \left(\sum_{j=1}^{t-1}(\tilde{\alpha}_{t,j} - \beta_{t,j})\,\tilde{x}_j\right)W_V W_O$$

Thus, $\|\tilde{a}_t - b_t\|_2$ can be bounded as

$$\|\tilde{a}_t - b_t\|_2 \le \lambda_V\lambda_O \sum_{j=1}^{t-1}(\tilde{\alpha}_{t,j} - \beta_{t,j})\|\tilde{x}_j\|_2 = \lambda_V\lambda_O \sum_{j=1}^{t-1}(\tilde{\alpha}_{t,j} - \beta_{t,j})$$

since $\|\tilde{x}_j\|_2 = 1$ for all $j \in [t]$. Now we analyze $\tilde{\alpha}_{t,j} - \beta_{t,j}$. Let $\hat{S}_t = S_t \setminus \{t\}$. Then $m_{k,j} = 1$ if and only if $j \in \hat{S}_t$. For convenience, let $r_{t,j} = 1/t \cdot \tilde{x}_t W_Q W_K^\top \tilde{x}_j^\top$. Thus, $\beta$ can be written as

$$\beta_{t,j} = \frac{\exp\left(r_{t,j}\right)}{\sum_{i\in\hat{S}_t}\exp\left(r_{t,i}\right) + \sum_{i\notin\hat{S}_t}\exp\left(r_{t,i}\right)}$$

Furthermore, for all $j \notin \hat{S}_t$, we have $\tilde{\alpha}_{t,j} = 0$. For all $j \in \hat{S}_t$, we have

$$\tilde{\alpha}_{t,j} = \frac{\exp\left(r_{t,j}\right)}{\sum_{i\in\hat{S}_t}\exp\left(r_{t,i}\right)}$$

Therefore, for all $j \in \hat{S}_t$, we have

$$\beta_{t,j} - \tilde{\alpha}_{t,j} = \exp\left(r_{t,j}\right) \cdot \frac{\sum_{i \notin \hat{S}_t} \exp\left(r_{t,i}\right)}{\left(\sum_{i \in \hat{S}_t} \exp\left(r_{t,i}\right)\right)\left(\sum_{i \in \hat{S}_t} \exp\left(r_{t,i}\right) + \sum_{i \notin \hat{S}_t} \exp\left(r_{t,i}\right)\right)}$$

$$= \frac{\exp\left(r_{t,j}\right)}{\sum_{i \in \hat{S}_t} \exp\left(r_{t,i}\right)} \cdot \frac{\sum_{i \notin \hat{S}_t} \exp\left(r_{t,i}\right)}{\sum_{i \in \hat{S}_t} \exp\left(r_{t,i}\right) + \sum_{i \notin \hat{S}_t} \exp\left(r_{t,i}\right)}$$

$$= \tilde{\alpha}_{t,j} \sum_{i \notin \hat{S}_t} \beta_{t,j}$$

Therefore, the bound of $\|\tilde{a}_t - b_t\|_2$ can be written as

$$\|\tilde{a}_t - b_t\|_2 \leq \lambda_V \lambda_O \left(\sum_{j \in \hat{S}_t}^{t-1} \tilde{\alpha}_{t,j} \sum_{i \notin \hat{S}_t} \beta_{t,j} - \sum_{j \notin \hat{S}_t} \beta_{t,j}\right) = 2\lambda_V \lambda_O \sum_{j \notin \hat{S}_t} \beta_{t,j}$$

where the last equality follows from $\sum_{j \in \hat{S}_t} \tilde{\alpha}_{t,j} = 1$. $\qquad\square$

Our last lemma shows the Lipschitzness of the MLP in (6).

**Lemma B.5.** *Let $\lambda_1, \lambda_2$ denote the largest singular values of $W_1, W_2$ in (6). For all $x_1, x_2 \in \mathbb{R}^d$, we have*

$$\|\mathcal{F}(x_1) - \mathcal{F}(x_2)\| \leq (1 + \lambda_1 \lambda_2) \|x_1 - x_2\|_2$$

*Proof.* Direct computation yields

$$\|\mathcal{F}(x_1) - \mathcal{F}(x_2)\| = \|(x_1 + W_2 \texttt{relu}(W_1 x_1)) - (x_2 + W_2 \texttt{relu}(W_1 x_2))\|$$

$$\leq \|x_1 - x_2\|_2 + \|W_2 \texttt{relu}(W_1 x_1) - W_2 \texttt{relu}(W_1 x_2)\|$$

$$\leq \|x_1 - x_2\|_2 + \lambda_2 \|\texttt{relu}(W_1 x_1) - \texttt{relu}(W_1 x_2)\|$$

$$\leq \|x_1 - x_2\|_2 + \lambda_2 \|W_1 (x_1 - x_2)\|_2$$

$$\leq \|x_1 - x_2\|_2 + \lambda_2 \lambda_1 \|x_1 - x_2\|_2$$

$$= (1 + \lambda_1 \lambda_2) \|x_1 - x_2\|_2$$

where in the third inequality we use the fact that $\texttt{relu}(\cdot)$ is 1-Lipschitz. $\qquad\square$

Now we turn to the proof of our main theorem. Combining all of the results, we have

$$a_t - \tilde{a}_t = \left(\sum_{j=1}^{t-1} \alpha_{t,j} x_j\right) W_V W_O - \left(\sum_{j=1}^{t-1} \tilde{\alpha}_{t,j} \tilde{x}_j\right) W_V W_O$$

$$= \underbrace{\left(\sum_{j=1}^{t-1} \alpha_{t,j} x_j\right) W_V W_O - \left(\sum_{j=1}^{t-1} \alpha_{t,j} \tilde{x}_j\right) W_V W_O}_{\mathcal{T}_1}$$

$$+ \underbrace{\left(\sum_{j=1}^{t-1} \alpha_{t,j} \tilde{x}_j\right) W_V W_O - \left(\sum_{j=1}^{t-1} \beta_{t,j} \tilde{x}_j\right) W_V W_O}_{\mathcal{T}_2}$$

$$+ \underbrace{\left(\sum_{j=1}^{t-1} \beta_{t,j} \tilde{x}_j\right) W_V W_O - \left(\sum_{j=1}^{t-1} \tilde{\alpha}_{t,j} \tilde{x}_j\right) W_V W_O}_{\mathcal{T}_3}$$

Therefore, by triangle inequality, we have

$$\|a_t - \tilde{a}_t\|_2 \leq \|\mathcal{T}_1\|_2 + \|\mathcal{T}_2\|_2 + \|\mathcal{T}_3\|_2 \tag{10}$$

To start, the magnitude of $\mathcal{T}_1$ can be bounded as

$$\|\mathcal{T}_1\|_2 = \left\|\left(\sum_{j=1}^{t-1}\alpha_{t,j}(x_{t,j}-\tilde{x}_{t,j})\right)W_V W_O\right\|_2$$

$$\leq \lambda_V \lambda_O \left\|\sum_{j=1}^{t-1}\alpha_{t,j}(x_{t,j}-\tilde{x}_{t,j})\right\|$$

$$\leq \lambda_V \lambda_O \sum_{j=1}^{t-1}\alpha_{t,j}\|x_{t,j}-\tilde{x}_{t,j}\|_2$$

$$\leq \lambda_V \lambda_O \Delta_t \sum_{j=1}^{t-1}\alpha_{t,j}$$

$$= \lambda_V \lambda_O \Delta_t$$

where in the third inequality we use $\|x_{t,j}-\tilde{x}_{t,j}\|_2 = \Delta_t$ and in the last equality we use $\sum_{j=1}^{t-1}\alpha_{t,j} = 1$. To bound the magnitude of $\mathcal{T}_2$, we apply Lemma B.3, which shows that $\|\alpha_t - \beta_t\| \leq 2\frac{\sqrt{t-1}}{t}\lambda_Q\lambda_K\Delta_t$ to get that

$$\|\mathcal{T}_2\|_2 = \left\|\left(\sum_{j=0}^{t-1}(\alpha_{t,j}-\beta_{t,j})\tilde{x}_j\right)W_V W_O\right\|_2$$

$$\leq \lambda_V \lambda_O \left\|\left(\sum_{j=0}^{t-1}(\alpha_{t,j}-\beta_{t,j})\tilde{x}_j\right)\right\|_2$$

$$\leq \lambda_V \lambda_O \sum_{j=0}^{t-1}|\alpha_{t,j}-\beta_{t,j}|\,\|\tilde{x}_j\|_2$$

$$\leq \lambda_V \lambda_O \|\alpha_t - \beta_t\|_1$$

$$\leq \sqrt{t-1}\lambda_V \lambda_O \|\alpha_t - \beta_t\|_2$$

$$\leq 2\left(1-\frac{1}{t}\right)\lambda_Q\lambda_K\lambda_V\lambda_O\Delta_t$$

Lastly, to bound the magnitude of $\mathcal{T}_3$, we use Lemma B.4 to get that

$$\|\mathcal{T}_3\|_2 \leq 2\lambda_V\lambda_O\sum_{j\notin\hat{S}_t}\beta_{t,j}$$

Putting things together for (10), we have

$$\|a_t - \tilde{a}_t\|_2 \leq \lambda_V\lambda_O\left(2\sum_{j\notin\hat{S}_t}\beta_{t,j}+(2\lambda_Q\lambda_K+1)\Delta_t\right)$$

By Lemma B.5 we can further show that

$$\|x_{t+1}-\tilde{x}_{t+1}\|_2 \leq (1+\lambda_1\lambda_2)\lambda_V\lambda_O\left(2\sum_{j\notin\hat{S}_t}\beta_{t,j}+(2\lambda_Q\lambda_K+1)\Delta_t\right)$$

By Theorem B.3, we have that with probability at least $1 - T_{\max}\exp\left(-\frac{\epsilon^2 b^2(T_{\min}-1)}{(k-2)^2(u-b)^2}\right) - T_{\max}\exp\left(-\frac{2(T_{\min}-1)(1-B/T_{\max})^2}{(1-\epsilon)^2}\right)$, it holds for all $t \in [T_{\min}, T_{\max}]$ that

$$\mathbb{E}\left[\sum_{j\notin\hat{S}_t}\beta_{t,j}\right] \leq \frac{(1-B/T_{\max})}{0.98(1-\epsilon)^2}\left(k-(k-1)\left(\frac{1-\epsilon}{B/T_{\max}-\epsilon}\right)^{\frac{1}{k-1}}\right) := \Delta_{\max}$$

Given that $\mathbb{E}\left[\|x_t - \tilde{x}_t\|\right] \leq 2\Delta_{\max}$, we have

$$\mathbb{E}\left[\|x_{t+1} - \tilde{x}_{t+1}\|_2\right] \leq (1 + \lambda_1\lambda_2)\lambda_V\lambda_O\left(2\Delta_{\max} + 2\left(2\lambda_Q\lambda_K + 1\right)\Delta_{\max}\right)$$
$$\leq 4\lambda_V\lambda_O(1 + \lambda_1\lambda_2)(1 + \lambda_Q\lambda_K)\Delta_{\max}$$

Thus, as long as $\lambda_V\lambda_O(1 + \lambda_1\lambda_2)(1 + \lambda_Q\lambda_K) \leq \frac{1}{2}$, we can guarantee that

$$\mathbb{E}\left[\|x_{t+1} - \tilde{x}_{t+1}\|_2\right] \leq 2\Delta_{\max}$$

Thus, for all $t \in [T_{\min}, T_{\max}]$, we have that

$$\mathbb{E}\left[\|x_t - \tilde{x}_t\|_2\right] \leq \frac{2.1(1 - B/T_{\max})}{(1 - \epsilon)^2}\left(k - (k - 1)\left(\frac{1 - \epsilon}{B/T_{\max} - \epsilon}\right)^{\frac{1}{k-1}}\right)$$

## B.3   Budgeted Cache

**Theorem B.3.** *Let $\beta_{t,j}$ be sampled from some power-law distribution $f(x) = c(x+b)^{-\gamma}$ with support on $[0, u - b)$ for some $k > 2$ and $u \geq 5b$. Let $S_t$ be defined in Theorem B.2, and define $\hat{S}_t = S_t \setminus \{t\}$. Then with probability at least $1 - T_{\max}\exp\left(-\frac{\epsilon^2 b^2(T_{\min}-1)}{(k-2)^2(u-b)^2}\right) - T_{\max}\exp\left(-\frac{2(T_{\min}-1)(1-B)^2}{(1-\epsilon)^2}\right)$ it holds for all $t \in T$ that*

$$\mathbb{E}\left[\sum_{j \notin \hat{S}_t}\beta_{t,j}\right] \leq \frac{(1 - B/T_{\max})}{0.98(1 - \epsilon)^2}\left(k - (k - 1)\left(\frac{1 - \epsilon}{B/T_{\max} - \epsilon}\right)^{\frac{1}{k-1}}\right) \tag{11}$$

We consider the case of maintaining a budget of $B$ by dropping the smallest $\beta_{t,j}$'s. Assume that $v_j$ has pdf $f(x) = c(x + b)^{-k}$ with support on $[0, u - b)$. To make things precise, we first compute $c$

$$c = \left(\int_0^{u-b}(x + b)^{-k}dx\right)^{-1} = \frac{k - 1}{b^{1-k} - u^{1-k}}$$

To start, we notice that

$$\int x(x + b)^{-k} = -\frac{(x + b)^{1-k}((k - 1)x + b)}{(k - 1)(k - 2)} := g(x)$$

Let $C = \sum_{j=1}^{t-1} v_j$, then the expectation of $C$ is

$$\mathbb{E}[C] = (t - 1)\mathbb{E}[v_1] = (t - 1)\frac{k - 1}{b^{1-k} - u^{1-k}}\int_0^\infty x(x + b)^{-k}dx$$

$$= (t - 1)\frac{k - 1}{b^{1-k} - u^{1-k}}(g(u) - g(0))$$

$$= (t - 1)\frac{k - 1}{b^{1-k} - u^{1-k}}\left(\frac{b^{2-k}}{(k - 1)(k - 2)} - \frac{u^{1-k}((k - 1)u - (k - 2)b)}{(k - 1)(k - 2)}\right)$$

$$= \frac{t - 1}{k - 2} \cdot \frac{b^{2-k} - (k - 1)u^{2-k} + (k - 2)bu^{1-k}}{b^{1-k} - u^{1-k}}$$

Let $\Delta = \frac{b^{2-k} - (k-1)u^{2-k} + (k-2)bu^{1-k}}{b^{1-k} - u^{1-k}}$. By Hoeffding's inequality, we have that

$$\mathbb{P}\left(C \leq (1 - \epsilon)\mathbb{E}[C]\right) \leq \exp\left(-\frac{2\epsilon^2\mathbb{E}[C]^2}{(t - 1)(u - b)^2}\right)$$

This implies that with probability at least $1 - \exp\left(-\frac{2\epsilon^2\Delta^2(t-1)}{(k-2)^2(u-b)^2}\right)$ we have

$$C \geq (1 - \epsilon)\Delta\frac{t - 1}{k - 2}$$

Now, we proceed to bound $\sum_{j \notin \hat{S}_t} \beta_{t,j}$ where $\hat{S}_t = \{j \in [t-1] : \beta_{t,j} \geq \frac{\gamma}{C}\}$. Equivalently, we can bound $C^{-1} \sum_{j=1}^{t-1} \mathbb{I}\{v_j \leq \gamma\} v_j$. Its expectation is given by

$$
\mathbb{E}\left[C^{-1} \sum_{j=1}^{t-1} \mathbb{I}\{v_j \leq \gamma\} v_j\right] \leq \frac{k-2}{(t-1)\Delta(1-\epsilon)} \mathbb{E}\left[\sum_{j=1}^{t-1} \mathbb{I}\{v_j \leq \gamma\} v_j\right]
$$

$$
= \frac{k-2}{\Delta(1-\epsilon)} \cdot \frac{k-1}{b^{1-k} - u^{1-k}} \int_0^\gamma x(x+b)^{-k} dx
$$

$$
= \frac{(k-1)(k-2)}{\Delta(1-\epsilon)\left(b^{1-k} - u^{1-k}\right)} \left(g(\gamma) - g(0)\right)
$$

We pause here and study how small can we choose $\gamma$. Notice that

$$
\mathbb{E}\left[\sum_{j=1}^{t-1} \mathbb{I}\{v_j \leq \gamma\}\right] = (t-1)\mathbb{P}(v_j \leq \gamma) = (t-1) \cdot \frac{b^{1-k} - (\gamma+b)^{1-k}}{b^{1-k} - u^{1-k}}
$$

By Hoeffding's inequality again, we have that

$$
\mathbb{P}\left(\sum_{j=1}^{t-1} \mathbb{I}\{v_j \leq \gamma\} \geq (1-\epsilon)(t-1) \cdot \frac{b^{1-k} - (\gamma+b)^{1-k}}{b^{1-k} - u^{1-k}}\right)
$$

$$
\leq \exp\left(-\frac{2(t-1)\epsilon^2 \left(b^{1-k} - (\gamma+b)^{1-k}\right)^2}{\left(b^{1-k} - u^{1-k}\right)^2}\right)
$$

Enforcing $\sum_{j=1}^{t-1} \mathbb{I}\{v_j \leq \gamma\} \geq T_{\max} - B$ gives $(\gamma+b)^{1-k} \leq b^{1-k} - \frac{1 - B/T_{\max}}{1-\epsilon}(b^{1-k} - u^{1-k})$, which can be satisfied as long as $\gamma \geq \left(\left(\frac{B/T_{\max} - \epsilon}{1-\epsilon}\right)^{\frac{1}{1-k}} - 1\right) b$. Therefore

$$
g(\gamma) = -\left(b^{1-k} - \frac{1 - B/T_{\max}}{1-\epsilon}(b^{1-k} - u^{1-k})\right) \frac{b + (k-1)\gamma}{(k-1)(k-2)}
$$

We further notice that

$$
b^{1-k} - \frac{1 - B/T_{\max}}{1-\epsilon}(b^{1-k} - u^{1-k}) \geq \frac{B/T_{\max} - \epsilon}{1-\epsilon}(b^{1-k} - u^{1-k})
$$

This gives

$$
\mathbb{E}\left[C^{-1} \sum_{j=1}^{t-1} \mathbb{I}\{v_j \leq \gamma\} v_j\right] \leq \frac{b(1 - B/T_{\max})}{\Delta(1-\epsilon)^2} - \frac{(k-1)(B/T_{\max} - \epsilon)\gamma}{\Delta(1-\epsilon)^2}
$$

$$
\leq \frac{b(1 - B/T_{\max})}{\Delta(1-\epsilon)^2} \left(k - (k-1)\left(\frac{1-\epsilon}{B/T_{\max} - \epsilon}\right)^{\frac{1}{k-1}}\right)
$$

Notice that if $u \geq 5b$, we have

$$
\Delta = b - (k-1)\left(\frac{u}{b}\right)^{1-k} \cdot \frac{b-u}{b^{1-k} - u^{1-k}} \leq 0.98b
$$

Therefore

$$
\mathbb{E}\left[C^{-1} \sum_{j=1}^{t-1} \mathbb{I}\{v_j \leq \gamma\} v_j\right] \leq \frac{(1 - B/T_{\max})}{0.98(1-\epsilon)^2} \left(k - (k-1)\left(\frac{1-\epsilon}{B/T_{\max} - \epsilon}\right)^{\frac{1}{k-1}}\right)
$$

holds with probability at least $1 - \exp\left(-\frac{\epsilon^2 b^2 (t-1)}{(k-2)^2 (u-b)^2}\right) - \exp\left(-\frac{2(t-1)(1 - B/T_{\max})^2}{(1-\epsilon)^2}\right)$. Taking a union bound gives the desired result.

