# OpenReview forum: "Scissorhands: Exploiting the Persistence of Importance Hypothesis for LLM KV Cache Compression at Test Time"
_NeurIPS.cc/2023/Conference — NeurIPS 2023 poster_

### Official Review · Reviewer_H8Ve · 2023-06-30

**Soundness:** 3 good
**Presentation:** 2 fair
**Contribution:** 3 good
**Rating:** 7
**Confidence:** 4

**Summary:**

The paper proposes a method for reducing the memory requirement of inference for large autoregressive language models.
The proposed techniques is based on the empirical observation that the high attention score tokens in a transformer self-attention layer maintain a remarkable persistence for different token positions in the input sequence.
This observation is used to consider a reduction of the memory required in the attention KV cache, by retaining only the information corresponding to 'pivotal tokens', which are defined as the token positions that have a high attention scores for some of the previous steps in the self-attention autoregressive generation.

**Strengths:**

The paper addresses the important issue of the high memory cost of inference for autoregressive generative language models, associated with the size of the KV cache, which has directly affects the inference execution time for large language models.
To the best of my knowledge, the observation of the persistence of the self-attention pattern is original, and gives a new angle in the understanding of this very important compute block.
The proposed method makes use of this observation to try to reduce the memory requirements associated with the KV cache of the self-attention, without requiring fine-tuning of the language model.

**Weaknesses:**

The exposition of the paper should be improved. In particular:

-- The authors make the claim that their technique "reduces the inference memory usage of the KV cache by up to 5x without compromising model quality", while from the results of Section 5 this is not achieved in all cases, and depending on the downstream task the performance is not maintained for compression larger than 2x or 3x.

-- The exposition of Algorithm 2 should be improved, adding further explanation and clarification in the text.

Other comments:

-- Line 91 and line 93: the sequence length is denoted by $p$, while the length of the prompt sequence was defined as $s$ at line 87.

-- Line 185: in the case of a single attention head, the matrices $W_Q$, $W_K$, $W_V$ should be of size $d \times d$ (and not $d \times p$), and the matrix $W_O$ of size $d \times d$ (and not $p \times d$).

-- In the equation after line 215 and in the equation after line 238, on both numerator and denominator of the softmax formula, $x_t W_K$ should be changed to $x_t W_Q$.

Minor comments:

-- Lines 67 and 69: "between who they ..." should be "between which tokens they ...".

-- Caption of Table 2: "This table summarises the maximum batch size ..." should be simply "Maximum batch size ...".

-- Line 102: "parallel strategy" should be "parallelism strategy".

-- Line 109: "Table 2.1" should be "Table 2".

-- Caption of Figure 2: The text "In this figure, we plot the persistence ratio and the corresponding size ..." should be changed to simply "Persistence ratio and corresponding size ...".

-- Line 132: "drawn from C4 [22]" may be rephrased as "drawn from the Colossal Clean Crawled Corpus (C4) [22]".

-- Algorithm 2: $[t-W,t]$ should be $[t-w,t]$ (the History Window Size is defined as lower-case $w$).

-- Line 263: "tokes" should be "tokens".

-- Caption of Figure 3: The text "This figure shows the accuracy trend of ..." should be changed to "Accuracy trend of ...".

**Questions:**

-- Regarding the persistence of the attention pattern, it seems that its manifestation critically depends on the cosine similarity between input and output of the MLP of equation (2) - could the authors elaborate on this point?

-- Appendix A.2 gives empirical evidence of large cosine similarity between input and output of the MLP block. In which setup have these results been obtained? How generally can this be assumed for different downstream tasks?

-- Section 6 provide evidence that a persistent attention pattern does not exist in randomly initialised models. Is my understanding correct that the current study was limited to inference, and that the authors were not in a position to explore the evolution of the attention map during training, even for smaller models?

**Limitations:**

The authors have acknowledged that running large language models can correspond to significant power consumption and CO2 emissions, but their work is aimed at improving the power efficiency of inference for large language models.

---

> ### Author Rebuttal · Authors · 2023-08-09
>
> We appreciate the reviewer’s supportive comments on novelty and constructive suggestions on the expositions. We have updated our manuscript and provided further details below. We hope our detailed clarification will clear the doubt about the significance and position of our work.
>
> **Q1: The exposition of the paper should be improved.**
>
> We appreciate your suggestions on the narratives. We have updated our manuscript accordingly. For example, (i) “reduces the inference memory usage of the KV cache 2-5x without compromising model quality”. (ii) a detailed explanation of Algorithm 2 is added in Section 4.2. “When the KV cache buffer is full, Scissorhands drops tokens from the memory according to Algorithms 2. The importance record is a counter that indicates how many times a token is deemed non-important. The counter for all tokens in the recent window size is set to 0. A higher counter indicates dropping from the cache. ”
>
> **Q2: Regarding the persistence of the attention pattern, is my understanding correct that its manifestation critically depends on the cosine similarity between the input and output of the MLP of equation (2)?**
>
> Our theoretical justification in Theorem 3.1 requires the input-output alignment of the MLP, but this is likely due to the internal difficulty of theoretically analyzing the property of trained weights in LLMs.  We didn’t explicitly design experiments to verify the causal relationship between the input-output alignment of the MLP and the persistence of the attention pattern due to the difficulties in controlling input-output alignment without access to training. We believe that it remains an open question to study the dependence of the persistence of the attention on the input-output alignment of the MLP, both theoretically and empirically.
>
> **Q3:  Appendix A.2 gives empirical evidence of large cosine similarity between the input and output of the MLP block. In which setup have these results been obtained? How generally can this be assumed for different downstream tasks?**
>
> We run OPT family model on 1000 sequences randomly drawn from the C4 validation dataset. We record the Cosine similarity between the input of the MLP block and the output of the MLP block for every transformer layer. We observe a similar trend across the entire OPT family. Further, recent work also observes similar phenomena. [1] shows in Figure 5 that the input and output of both the MLP block and attention block have high cosine similarity.
>
> [1] Liu, Zichang, et al. "Deja vu: Contextual sparsity for efficient llms at inference time." International Conference on Machine Learning. PMLR, 2023.
>
> **Q4: Section 6 provides evidence that a persistent attention pattern does not exist in randomly initialized models. Is my understanding correct that the current study was limited to inference and that the authors were not in a position to explore the evolution of the attention map during training, even for smaller models?**
>
> The current work focuses on the inference stage. Unfortunately, we don’t have access to pre-train a large language model. Actually, the intention of comparing with a randomly initialized model is to spark the interest of the community to investigate the training evolution of attention maps. Investigating such phenomena during training may lead to understanding model behaviors and intuition for designing sparse attention.

---

> > ### Comment · Reviewer_H8Ve · 2023-08-16
> >
> > I would like to thank the authors for addressing the points raised in my review and for proposing to clarify the exposition.
> >
> > As an additional comment, I have noticed that line 26 of the Introduction gives a KV cache memory requirement of around 950 GB for the OPT-175B model with batch size 128 and sequence length 2048, while this should be around 1200 GB, as also given in Table 1.

---

> > > ### Comment · Reviewer_H8Ve · 2023-08-19
> > >
> > > I have now considered the authors's rebuttal and the other reviewers' comments, and I appreciate the additional experimental results provided by the authors.
> > >
> > > It is also great that the authors have taken into account the reviewers' comments to improve the exposition of the original manuscript.
> > >
> > > Considering the basic contribution of the paper to be important both for the practical implications and for the understanding of autoregressive inference, in the light of the additional experimental evaluation I have now increased the score of my original review.

---

> > > > ### Author Response · Authors · 2023-08-21
> > > > **Response**
> > > >
> > > > Thanks for recognizing the significance of our work. Also, we want to thank the reviewer's effort and time in the reviewing process to help us improve our paper.

---

### Official Review · Reviewer_AWSm · 2023-07-07

**Soundness:** 4 excellent
**Presentation:** 4 excellent
**Contribution:** 4 excellent
**Rating:** 7
**Confidence:** 5

**Summary:**

This work studies attention patterns of LLMs and find that high-valued token-token patterns repeat for particular pivotal tokens. Since these patterns repeat, they can be compressed by storing only the pivotal tokens thus reducing the KV-cache size during inference.

Recommendation:

I recommend acceptance. The paper has a novel important analysis, which is already highly valuable. The proposed KV-compression is robustly evaluated. I  have no concerns and recommend acceptance. However, since the impact of this work is in a limited sub-area (attention-specific research mostly), I would not highlight it at the conference.

**Strengths:**

- The analysis is thorough and insightful and will guide further development of KV-cache compression
- The experimental validation is convincing using both different model scales and different zero-shot tasks
- Results are excellent, making this approach practically useful

**Weaknesses:**

OPT has some unusual patterns, for example later layers have larger variances for hidden states. Other models do not have this property and are more stable. As such, I would imagine that the results would change slightly when considering a different model (likely the results would improve in later layer compared to the OPT results reported in the paper). It would be good to have these additional results for a different model (LLaMA would be best). But even without these results the paper is experimentally robust.

**Questions:**

The paper is very clear, and I have no questions.

**Limitations:**

Limitations are discussed.

---

> ### Author Rebuttal · Authors · 2023-08-09
>
> Thank the reviewer for the supportive comments, recognizing the novelty of our analysis, and thorough evaluations. We understand that another model family would further demonstrate the generality of our hypothesis and methods. However, we couldn't finish this within the time frame due to the time constraints of rebuttal and additional experiment efforts. We will try to update the manuscript with results on LLaMA later.

---

### Official Review · Reviewer_g2hZ · 2023-07-07

**Soundness:** 3 good
**Presentation:** 3 good
**Contribution:** 3 good
**Rating:** 5
**Confidence:** 3

**Summary:**

This work aims to increase the throughput of inference system by reducing the size of the KV cache.
- The KV cache comes from saving forward pass computations and can take up up to 5x the model size.

The paper proposes to discard part of the KV cache, based on the observation that the set of tokens that get high attention are largely shared across positions, which the paper calls the Persistence of Importance Hypothesis.
- The proposed method only keeps the most recent tokens and "pivotal tokens", which are tokens whose attention scores are higher than a threshold. Empirical results using OPT show - T
- Theoretical analyses (Theorem 4.1) show that such process doesn't affect the attention output much.

**Strengths:**

- The problem is well motivated and the solution makes intuitive sense.
- The empirical results are promising:
  - The proposed method achieves up to 5x compression and doesn't sacrifice much performance.
  - The proposed method is compatible with 4-bit quantization.


**Weaknesses:**

Some design decisions are not well justified, and there are insufficient baseline comparisons.

**Questions:**

- The main motivation of the paper is to improve throughput, and the paper has mostly discussed the improvement in terms of memory. What's the improvement on wall clock time?
- The current scheme of selecting pivotal tokens is attention-based. Are there other methods / baseline to compare to?
  - e.g. Keep the recent window as the current method, and change the selection of distance tokens based on word frequency?
  - e.g. Instead of thresholding the token importance, another way could be to select tokens that contributes to e.g. 90% attention weights, similar to Nucleus sampling; how do you think this would compare?
- Results in Fig 2: I'm not sure $|S_{0\rightarrow t}|$ can be considered as "considerably smaller than $t$" based on Fig 2 (b).
- Clarification on $\alpha = \frac{1}{t}$: for $S_{0 \rightarrow t}$, does then mean for $\tau \in [t]$, $S_{\tau}$ is calculated with $\frac{1}{\tau}$ or $\frac{1}{t}$?
- Equation 1: what is there a scaling of 1/t within the softmax?

Writing:
- The paragraph starting at line 202 could be made clearer.
- Line 238: $w,r$ have been defined in Alg 2; have they been defined in the main text?
- Minor point: Fig 3: might be better to use different line style / color for models / tasks.
- Please fix typos, e.g. line 202, 263, 295, 308.


**Limitations:**

There is no direct societal impact.

---

> ### Author Rebuttal · Authors · 2023-08-09
>
>
> We appreciate the reviewer’s thoughtful suggestions and supportive comments. We have updated our manuscript to provide further clarification and improve writing details.
>
> **Q1: The paper has mostly discussed the improvement in terms of memory. What's the improvement on wall clock time?**
>
> The main focus of our work is on memory usage. We agree that compressing the KV cache also influences inference efficiency, which is worthy of investigation. Our method saves the attention computation because dropping tokens from the KV cache effectively reduces the sequence length. On the other hand, the overhead we introduce stems from identifying which token to drop. The overhead is rather limited compared to the saving.
>
> We provide throughput measured in tokens per second in Table 8. We based the throughput measurement based on FlexGen[1]. We use a prompt length of 512 and a generation length of 1024. For all other settings, we keep the default from FlexGen. As shown in Table 8, we observe a slight improvement (1.2X ) in inference throughput. Due to the rebuttal time limitation, we could only implement a basic version. Implementation can be further improved to realize a significant gain in both memory usage and throughput.
>
> [1] Sheng, Ying, et al. "High-throughput generative inference of large language models with a single gpu." arXiv preprint arXiv:2303.06865 (2023).
>
>
> **Q2: The current scheme of selecting pivotal tokens is attention-based. Are there other methods/baselines to compare to?**
>
> The memory problem from the KV cache is rather a recent constraint. The most established direction is applying quantization to reduce the memory footprint. We believe quantization are orthogonal direction rather than a competing direction. We show the possibility of combining quantization and Scissorhands in Section 5.
>
> We understand that adding baselines would help understand Scisscorhands’s performance and limitations. Thus, we implement a straightforward position-based method: Local Window. Local Window only keeps the neighbor tokens’ cache. We take C4 and divide it into three subsets depending on the sequence length. C4-[256-512] contains data sequences that are longer than 256 tokens but less than 512 tokens. C4-[512-1024] contains data sequences that are longer than 512 tokens but less than 1024 tokens. C4-[1024-2048] contains data sequences that are longer than 1024 tokens but less than 2048 tokens. Results are summarized in Table 4.
>
> We observe the perplexity of the full cache model degrades slightly with the growing sequence length. At all sequence lengths, Scissorhands performance is comparable against the full cache model, while Local Window incurs a significant quality loss. This demonstrates that keeping the pivotal tokens is important to reserve model performance.
>
> It is also interesting to note that at longer sequence lengths, the local window has higher accuracy. This also shows at longer sequence length, the attention mechanism in current architecture tends to focus on recent context.
>
> **Q3: Clarification regarding to $|S_{0\rightarrow t}|$ and $\alpha = 1/t$.**
>
> $|S_{0\rightarrow t}| / t$ varies across layers. Using OPT-66B as an example, the early and later layers have a much lower ratio (around 0.2 on average), while the middle layer has a higher ratio, around 0.5 on average. For $S_{0 \rightarrow t}$, $\tau \in [t]$, $S_{\tau}$ is calculated with $\frac{1}{\tau}$. The main message of this observation is to show that only a subset of tokens will carry its influence further. We have updated the manuscript to make sure the descriptions are accurate and more in detail.
>
> **Q4: Equation 1 what is there a scaling of 1/t within the softmax?**
>
> We added the $\frac{1}{t}$ scaling in the softmax to facilitate the theoretical analysis in Section 4.2. In particular, since the size of the not subsample W_Q and W_K grows along the token generation process, we need this scaling to control the error propagation through the softmax. Alternatively, we can also use the $\frac{1}{\sqrt{d}}$ scaling and add an assumption to enforce that the maximum sequence generation length satisfies $T \leq \sqrt{d}$.

---

> > ### Comment · Reviewer_g2hZ · 2023-08-13
> > **Follow-up questions**
> >
> > Thank you for the clarifications and the additional evaluation on the throughput!
> >
> > My concern remains that even though the idea of sparsely sub-selecting tokens is interesting, some design choices seem a bit arbitrary and that there need to be more comparison / ablation.
> > - Regarding Q2: Thank you for the experiments, though I'm not sure using a Local Window alone would be a competitive baseline.
> >   - I didn't mean to use local window only since it clearly will lose information in the distant past; I was thinking about using local window in combination with other ways to select distant tokens; e.g. non-context-based selection, or selection based on the similarity to the attention weight matrices as Deja Vu did.
> >   - I'm a bit concerned about attention-score-based selection is not ideal, since attention scores don't translate directly to "importance" due to the presence of other components e.g. $W_V$. For example, could you explain why you chose to use attention score (i.e. softmax output) instead of e.g. the norm of $a_t$ (i.e. taking consideration $W_V, W_O$, as defined in eq (1))? It seems that in Theorem 3.1, the assumptions also take into account $W_V, W_O$.
> > - Regarding Q3: choosing the threshold to be $1/t$ seems arbitrary and need more comparison, e.g. comparing with adaptive threshold such as selecting the top-$k$ candidates.
> >
> > The results in Table 7 are interesting. The current example seems to suggest that a higher compression rate on the KV cache results in shorter generation. Is this true in general?

---

> > > ### Author Response · Authors · 2023-08-18
> > > **Reply to further questions**
> > >
> > > We definitely agree with you that there could always be more design choices to explore. There are around two main questions, (1) What should we use as the pivotal score, and (2) what are the best criteria for a pivotal token given the score?
> > >
> > > For the first question, there are multiple choices ( we really appreciated your suggestion here). One could consider the softmax score, the norm after $W_v$. We think the norm after $W_o$ is less promising because it will lose the head granularity. I.e. We can’t drop different tokens for different heads. The norm after $W_v$ are mainly from an implementation perspective. Most model implementations use a batch matrix-matrix product such that the output after $W_v$ is summarized along the sequence length. And to use the norm after $W_v$ would require changing to matrix multiplication and then summarizing, while bmm is typically a more efficient operation in both memory and latency; thus, we went with the simpler attention score. Nevertheless, we agree that it is definitely interesting to see from the accuracy perspective what other information can be used as an important measurement and we would leave this as future work.
> > >
> > > For the second question, there are two main choices: static thresholding and adaptive thresholding. With adaptive thresholding, one could use the top-K value as the threshold, as the reviewer suggested. We update Table 4 to provide a comparison of the C4 perplexity. The TopK threshold shows promising performance as well, but we didn’t observe a performance gap compared with the static threshold $1/t$. One potential reason is the sort operation over the history window. We also checked that there is a really high overlapping ratio between the token from the TopK threshold and the static thresholding.
> > >
> > >
> > > |                                | [256-512] | [512-1024] | [1024-2048] |
> > > |--------------------------------|-----------|------------|-------------|
> > > | OPT13B                         | 8.7968    | 9.1017     | 9.3005      |
> > > | OPT13B + static thresholding   | 8.7972    | 9.1011     | 9.3009      |
> > > | OPT13B + adaptive thresholding | 8.7954    | 9.1032     | 9.3012      |
> > >
> > >
> > > Regarding the generation length, we looked at the average generation length at different compression ratios, and we couldn’t draw a conclusive argument at the moment.

---

> > > > ### Comment · Reviewer_g2hZ · 2023-08-21
> > > >
> > > > Thank you for the further clarifications and results!
> > > >
> > > > For "what to use as a pivotal score", I'd like to clarify that my concern of using the attention scores directly is that this seems to rely on the assumption that attention scores reflect some notion of importance, which however may not be true [e.g. 1,2].
> > > >
> > > > [1] Sarthak Jain, Byron C. Wallace. Attention is not Explanation.
> > > >
> > > > [2] Sofia Serrano, Noah A. Smith. Is Attention Interpretable?

---

### Official Review · Reviewer_UGwp · 2023-07-11

**Soundness:** 1 poor
**Presentation:** 2 fair
**Contribution:** 1 poor
**Rating:** 3
**Confidence:** 5

**Summary:**

The paper introduces a new way to compress KV cache for LLM serving. It showed the proposed approach can reduce the memory usage for inference up to 5 times. However, there is no evaluation on the performance on the generative datasets to justify the claim that there is no loss in the serving accuracy.

**Strengths:**

Ideas and math are well written. The direction of using pivotal tokens could be interesting.


**Weaknesses:**

The authors need to do a better job in evaluation to justify the contributions.


1. There is no benchmarks on the inference efficiency (latency and throughput) with the new approach. Was inference efficiency affected by the compression ratio?

2. The motivation of the method is to improve serving, especially allowing larger batch size for online serving of LLMs. However, the tested datasets are classification tasks, which only required one single forward pass without involving the decoding loop. We don't know how this method affects the performance of generative datasets (which involved decoding loops), like those question & answers ones as well as the instruction tuning ones.

3. KV state is large only when sequence length is very long. However, the tested dataset have a very short sequence length.

4. Please show the table with numbers instead of figures when claiming no loss in accuracy. In LLM evaluation, every single digit matters.


**Questions:**

Multi-query attention (K V states with only one head) already becomes commonly used for production LLMs. The KV state with MQA is  (1/num_heads) smaller than the one used for the old multi-head attention used in OPT and LLAMA. I suggests authors use MQA for the baseline.

https://arxiv.org/abs/1911.02150

**Limitations:**

The motivation of this approach is for efficiency deployment. However, the authors need to do a better job in showing that in the evaluation and experiments session. There was just a mismatch in the experimental setup (like choice of dataset and benchmarks) and the claims.

---

> ### Author Rebuttal · Authors · 2023-08-09
>
> We are glad that the reviewer finds our hypothesis and method interesting. We have tried to address your questions carefully. We hope the reviewer can consider raising your score in light of our response.
>
>
>
> **Q1: There are no benchmarks on the inference efficiency (latency and throughput) with the new approach. Was inference efficiency affected by the compression ratio?**
>
> The main focus of our work is on memory usage. We agree that compressing the KV cache also influences inference efficiency, which is worthy of investigation. Our method saves the attention computation because dropping tokens from the KV cache effectively reduces the sequence length. On the other hand, the overhead we introduce stems from identifying which token to drop. The overhead is rather limited compared to the saving.
>
>
>
> We provide throughput measured in tokens per second in Table 8. We based the throughput measurement based on FlexGen[1]. We use a prompt length of 512 and a generation length of 1024. For all other settings, we keep the default from FlexGen. As shown in Table 8, we observe a slight improvement (1.2X ) in inference throughput. Due to the rebuttal time limitation, we could only implement a basic version. Implementation can be further improved to realize a significant gain in both memory usage and throughput.
>
>
>
> [1] Sheng, Ying, et al. "High-throughput generative inference of large language models with a single gpu." arXiv preprint arXiv:2303.06865 (2023).
>
>
>
> **Q2: We don't know how this method affects the performance of generative datasets like those questions & answers ones as well as the instruction tuning ones. KV state is large only when the sequence length is very long. However, the tested dataset has a very short sequence length.**
>
>
>
> To better understand  Scissorhands’s performance and limitations, we performed three additional experiments based on the reviewer's suggestion. We first investigate sequence length’s influence on our method. Second, we provide generated texts at different compression ratios for more intuitive illustration. Third, we provide raw numbers on the tested dataset and results on three additional datasets.
>
> - We take C4 and divide it into three subsets depending on the sequence length. C4-[256-512] contains data sequences that are longer than 256 tokens but less than 512 tokens. C4-[512-1024] contains data sequences longer than 512 tokens but less than 1024 tokens. C4-[1024-2048] contains data sequences that are longer than 1024 tokens but less than 2048. We summarize the perplexity in Table 4. In addition to the full cache and Scissorhands, we implement Local Window. Local Window only keeps the neighbor tokens’ cache. All numbers are measured at 3X compression.
>
>
> 	We observe the perplexity of the full cache model degrades slightly with the growing sequence length. Scissorhands' performance is comparable to the full cache model at different sequence lengths. This is because Scissorhands could identify the pivotal tokens. On the contrary, there is a significant quality loss if one only keeps recent tokens.
>
> - We provide generation examples with OPT-13B. We compare the generation between full cache and Scissorhands at different compression ratios. The results are provided in Table 7.
>
>
> 	We observe exact same generation at 3X compression. At 6X compression, the generation starts to differ from the original model in the second sentence. Even though there is a degradation in generation quality at 6X compression, the generation is still coherent.
>
> - We set up our experiment datasets following the paper focusing on efficient inference [1][2][3]. We also test language modeling perplexity, as it is generally a good indicator of downstream task performance. We would also like to clarify that the computation during the prompting stage is calculated as if the key/value cache is dropped when obtaining accuracy.
>
>
> 	In Table 5, we provide accuracy numbers, as requested by the reviewer, for the datasets we used in the experiment section. In Table 6, we also provide results on three additional datasets. All accuracy is measured at 3X compression. Table 5 and Table 6 show that Sccissorhand provides comparable accuracy as the full cache model.
>
> [1] Frantar, Elias, et al. "Gptq: Accurate post-training quantization for generative pre-trained transformers." arXiv preprint arXiv:2210.17323 (2022).
>
> [2] Xiao, Guangxuan, et al. "Smoothquant: Accurate and efficient post-training quantization for large language models." International Conference on Machine Learning. PMLR, 2023.
>
> [3] Liu, Zichang, et al. "Deja vu: Contextual sparsity for efficient llms at inference time." International Conference on Machine Learning. PMLR, 2023.
>
>
>
> Q3: Multi-query attention is (1/num_heads) smaller than the one used for the old multi-head attention used in OPT and LLAMA. I suggest authors use MQA for the baseline.
>
> Thanks for the suggestion. MQA has indeed become a popular choice for production LLMs. However, adopting MQA requires training or finetuning the LLMs. Our work focuses on inference, where we are given trained LLMs.

---

### Official Review · Reviewer_h9xt · 2023-07-27

**Soundness:** 2 fair
**Presentation:** 3 good
**Contribution:** 3 good
**Rating:** 6
**Confidence:** 3

**Summary:**

This paper presents a new KV cache mechanism that uses an attention importance score to drop out tokens from the cache in transformer models. This allows up to 5x compression of the cache without degradation on assessed tasks.

**Strengths:**

This is likely to be useful to practictioners and researchers alike by making large model and long context inference more accessible on commodity GPUs. With appropriate discussion of limitations and caveating of claims, this should be a useful addition to the increasing toolkit for efficient use of large language models up to the 66B parameter scale.

**Weaknesses:**

+ The evaluation appears to focus on 4 datasets: HellaSwag, MathQA, PIQA, and Winograde. But I wonder whether the lack of performance degradation is a function of these particular datasets. For example, wouldn't a more adversarial dataset where there is an  fact at the beginning of the text that is unimportant until the very final token generations? By the time you need to re-use the fact, it will have dropped out of the cache. The paper would be improved with a more adversarial evaluation which shows the limitations of this approach to give users a more realistic picture of its limits. Or is it as the paper hints at in the conclusion, it doesn't matter because the architectural bias would prevent the attention mechanism from attending to those tokens anyways even if they aren't dropped out?
----
EDIT: Based on discussion with authors, conditioned on the inclusion of an experiment/discussion showing the limits of this approach, I have increased my score.

**Questions:**

+ Are there any other baselines you could compare against to give readers a reference point for determining whether to use this approach or another similarly situated approach?
+ The evaluation appears to focus on 4 datasets: HellaSwag, MathQA, PIQA, and Winograde. But I wonder whether the lack of performance degradation is a function of these particular datasets. For example, wouldn't a more adversarial dataset where there is an  fact at the beginning of the text that is unimportant until the very final token generations? By the time you need to re-use the fact, it will have dropped out of the cache. The paper would be improved with a more adversarial evaluation which shows the limitations of this approach to give users a more realistic picture of its limits. Or is it as the paper hints at in the conclusion, it doesn't matter because the architectural bias would prevent the attention mechanism from attending to those tokens anyways even if they aren't dropped out?

**Limitations:**

See above in weaknesses and questions.

---

> ### Author Rebuttal · Authors · 2023-08-09
>
> Thank the reviewer for the supportive comments on the significance of our work. We have carefully considered the reviewer’s constructive suggestions and updated the manuscript accordingly.
>
>
> **Q1: The evaluation appears to focus on 4 datasets: HellaSwag, MathQA, PIQA, and Winograd. But I wonder whether the lack of performance degradation is a function of these particular datasets. Are there any other baselines you could compare against to give readers a reference point for determining whether to use this approach or another similarly situated approach?**
>
>
>
> To better understand  Scissorhands’s performance and limitations, we performed three sets of additional experiments. We first investigate the performance with different sequence lengths. Second, we provide generated texts at different compression ratios for more intuitive illustration. Third, we provide results on three additional datasets.
>
>  - We take C4 and divide it into three subsets depending on the sequence length. C4-[256-512] contains data sequences that are longer than 256 tokens but less than 512 tokens. C4-[512-1024] contains data sequences that are longer than 512 tokens but less than 1024 tokens. C4-[1024-2048] contains data sequences that are longer than 1024 tokens but less than 2048 tokens. In addition to the full cache and Scissorhands, we implement Local Window. Local Window only keeps the neighbor tokens’ cache. All numbers are measured at 3X compression. Results are summarized in Table 4.
>
>
> 	We observe the perplexity of the full cache model degrades slightly with the growing sequence length. At all sequence lengths, Scissorhand's performance is comparable against the full cache model, while Local Window incurs a significant quality loss. This demonstrates that keeping the pivotal tokens is important to reserve model performance.
>
> 	It is also interesting to note that at longer sequence lengths, the local window has higher accuracy. This also shows at longer sequence lengths, the attention mechanism in current architecture tends to focus on recent context.
>
> - We provide generation examples with OPT-13B. We compare the generation between full cache and Scissorhands at different compression ratios. The results are provided in Table 7.
>
>
> 	We observe exact same generation at 3X compression. At 6X compression, the generation starts to differ from the original model in the second sentence. Even though there is a degradation in generation quality at 6X compression, the generation is still coherent.
>
> - Generally, language modeling perplexity is a good indicator for downstream tasks. Following a recent paper on LLM efficiency [1][2][3], we provide results on three additional datasets in Table 6. All accuracy is measured at 3X compression. Table 5 and Table 6 show that Sccissorhand provides comparable accuracy to the full cache model.
>
>
> [1] Frantar, Elias, et al. "Gptq: Accurate post-training quantization for generative pre-trained transformers." arXiv preprint arXiv:2210.17323 (2022).
>
> [2] Xiao, Guangxuan, et al. "Smoothquant: Accurate and efficient post-training quantization for large language models." International Conference on Machine Learning. PMLR, 2023.
>
> [3] Liu, Zichang, et al. "Deja vu: Contextual sparsity for efficient llms at inference time." International Conference on Machine Learning. PMLR, 2023.

---

> > ### Comment · Reviewer_h9xt · 2023-08-14
> > **Response**
> >
> > Thank you for providing these additional experiments. While I appreciate them, they do not quite address my concerns. The additional datasets (RTE, OpenBookQA, COPA) also don't really require long contexts. I was hoping to see something a bit more adversarial to give readers a picture of where this method might begin to falter. For example, what about an experiment with some sort of pseudocode like this:
> >
> > ```
> > import uuid
> >
> > for i in range(1000):
> >     secret_key = uuid.uuid4()
> >     test_string = f"Here is a private key: {secret_key}. {2000_filler_tokens_of_c4}. What was the private key?"
> >     output = llm(test_string)
> >     # Does the output from the llm match the secret key?
> > ```
> >
> > Such a task would require ignoring the 2000 tokens of filler content and recalling the private key from the beginning of the sequence. If the cache is sufficiently small, wouldn't the cache eviction method in this paper remove the uuid from the cache since it's not needed for the 2000 filler tokens (and therefore fail the task)?
> >
> > Of course, OPT isn't instruction tuned so I don't expect this to work out of the box. But I would expect some sort of experiment along these lines that pushes the heuristic cache eviction algorithm into a situation where we suspect it might be suboptimal so that we have a thorough characterization of the method. Of course this might be an unusual task that isn't typically benchmarked, but it nonetheless might be something that users encounter.
> >
> > I think it's totally okay for the method to fail such a test and still be a valuable contribution, but it's important to give the readers a thorough picture of the method.

---

> > > ### Author Response · Authors · 2023-08-18
> > > **Reply to the further question**
> > >
> > > This is a really interesting setup to understand the limitation under long-range skip dependency. We first want to thank you for this suggestion! We change the prompt to
> > >
> > >     “My dog name is Latte. He is a border Collie.” + sequence from c4 dataset + “My dog is ”
> > >
> > > We drop the KV cache for the prompt and start the generation process. And we provide the generation example below. We hope these generation examples, along with the experiments with different sequence lengths can provide a more thorough understanding of the method.
> > >
> > > > OPT-13B: a border collie.
> > > >
> > > > OPT-13B ~2X Compression: a border collie.
> > > >
> > > > OPT-13B ~3X Compression: a good boy.
> > >
> > > We tried this with a number of prompt we randomly selected from C4, and we noticed that for a large portion of them, OPT-13B dense itself fails to recall the name or the breed. We also notice that above around 2X compression of prompt KV cache would incur the loss of information. It is a bit tricky to evaluate in terms of accuracy, as any generation related to the name, the breed, or even the breed’s characteristics could be reasonable. It would be interesting to construct a benchmark dataset and evaluate this long-range skip dependency.

---

> > > > ### Comment · Reviewer_h9xt · 2023-08-18
> > > > **Response**
> > > >
> > > > Thank you for the response! I appreciate the additional qualitative experiment. Conditioned on the inclusion of some of this discussion in the final paper, along with a more quantitative version of the experiment accounting for variance, I am inclined to increase my initial review score.

---

> > > > > ### Author Response · Authors · 2023-08-21
> > > > > **Response**
> > > > >
> > > > > We appreciate the reviewer increasing the score. Reviewer's suggestion has greatly helped us improve our paper.  We thank the reviewer's time and effort spent on the reviewing process. We will make sure the additional experiments and discussion on included in the manuscript.

---

### Author Rebuttal · Authors · 2023-08-09

**Response to All Reviewers:**

We thank reviewers [R1(h9xt), R2(UGwp), R3(g2hZ), R4(AWSm), R5(H8Ve)] for their thoughtful and highly supportive feedback! We were glad that the reviewers found the problem significant and interesting [R1, R3, R5], the observations and theoretical analysis insightful and valuable [R2, R4, R5], the methods novel and clear [R2, R3, R4], and the experimental results convincing and promising [R3, R4, R5].



We have updated the paper to incorporate constructive suggestions, which will show in the revision. We summarize the attached additional experiments:

[ R1, R2, R5] Additional experiments and baselines (Table 4, 5, 6, 7):


-   We investigate the performance at different sequence lengths to understand the limitation.

-   We provide generated texts at different compression ratios for more intuitive illustration.

-   We provide results on three additional datasets.

[R2, R3] Inference efficiency (Table 8):


-   Scissorhands save attention computation because of the decreased sequence length from dropped KV cache. We show a 1.2X improvement in inference throughput.

---

### Decision · Program_Chairs · 2023-09-21

**Decision:**

Accept (poster)

**Comment:**

This paper proposes a method to reduce memory costs incurred from storing past keys and values during inference with large language models. It leverages the observation that only a few important tokens influence later predictions, and presents a method for identifying such tokens and pruning the rest from the cache. Experiments on generation tasks validate the memory savings with a limited performance decrease. Overall, all reviewers except for one agreed that the paper should be accepted. The rejecting reviewer was concerned about the short generation length of the tasks included in the initial submission, the lack of mention of speedups (instead of just memory savings), and the lack of comparison to multi-query attention (MQA). The authors provided additional experiments on longer-form generation, measured speedups (which were modest, but nontrivial), and pointed out that MQA involves additional training while their method does not. The rejecting reviewer did not provide a response to the rebuttal; I think an appropriate response would have been to raise their score given that all concerns were addressed. However, I do agree that the paper should at least mention MQA as a possible orthogonal way to obtain speedups.